# NMH-CS 3.0: a C# Programming Language and Windows System based Ecohydrological Model Derived from Noah-MP

Yong-He Liu[1], Zong-Liang Yang[2]

[1] School of Resources and Environment, Henan Polytechnic University, Jiaozuo, Henan, China

[2] Jackson School of Geoscience, University of Texas at Austin, Austin TX, USA

**Correspondence:** Yong-He Liu (yonghe_hpu@163.com)

**Abstract.** The community Noah with multi-parameterization options (Noah-MP) land surface model (LSM) is widely used in studies from uncoupled land surface hydrometeorology and ecohydrology to coupled weather and climate predictions. In this study, we developed NMH-CS 3.0, a hydrological model written in CSharp(C#). NMH-CS 3.0 is a new model developed by faithfully translating the FORTRAN version Noah-MP from the uncoupled WRF-Hydro 3.0, and is coupled with a river routing model. NMH-CS has the capacity of execution on Windows systems, utilizing the multi-core CPUs commonly available in today's personal computers. The code of NMH-CS has been tested to ensure that it produces a high-degree of consistency with the output of the original WRF-Hydro. High-resolution (6 km) simulations were conducted and assessed over a grid domain covering the entire Yellow River Basin and the most of North China. The spatial maps and temporal variations of many state variables simulated by NMH-CS 3.0 and WRF-Hydro/Noah-MP demonstrate highly consistent results, with occasionally minor discrepancies. The river discharge for the Yellow River simulated by the new model with various scheme combinations of six parameterizations exhibit general agreement with the natural river discharge at the Lanzhou station. NMH-CS can be regarded as a reliable replica of Noah-MP in WRF-Hydro 3.0, but it can leverage the modern, powerful, and user-friendly features brought by the C# language to significantly improve the efficiency of the model users and developers.

## 1 Introduction.

In contemporary hydrological prediction and flood warning applications, the effectiveness of hydrological models hinges on their ability to delineate intricate energy and water processes on the land surface, surpassing the capabilities of traditional rainfall-runoff models. To address this demand, certain land surface models (LSMs) utilized by atmospheric science communities have been bolstered with hydrological simulation features, as observed in WRF-Hydro (Lin et al., 2018), or conventional rainfall-runoff models have been enriched with more comprehensive descriptions of land surface processes, exemplified by the VIC model (Liang et al., 1994).

The Noah Land Surface Model with multi-parameterizations (Noah-MP) (Niu et al., 2011; Yang et al., 2011) stands out as a robust tool for studying global water issues, serving as the foundation for models like WRF-Hydro, which incorporates Noah-MP (Gochis, 2020). However, the code for Noah-MP and WRF-Hydro is written in FORTRAN, a 'legacy' language, posing challenges for code analysis and editing, unlike more modern languages such as CSharp(C#), because FORTRAN lacks the similar

intelligent efficient programming tools that are now common for C#. This limitation makes it arduous for users unfamiliar with FORTRAN to efficiently comprehend and modify the code. Additionally, Noah-MP and WRF-Hydro necessitate a UNIX-like operating system, causing inconvenience for the users and developers relying on Windows systems. Therefore, there is a compelling need to code Noah-MP in a contemporary modern programming language, to gain a wider accessibility of the Noah-MP model.

We designed NMH-CS 3.0, a Noah-MP based Hydrological model coded using the CSharp (C#) programming language. This model was crafted by creating a framework and accurately translating the original Noah-MP LSM code from WRF-Hydro 3.0 and coupling with a Muskingum method-based river routing model(Liu et al., 2023). C#, recognized for its modern and object-oriented approach, is widely used for software development across various platforms, particularly on the Windows operating system. According to the TIOBE Programming Community Index (https://www.tiobe.com/) for October 2024, C# ranks fifth among major programming languages with a user base of 5.6%, while Fortran ranks ninth with only 1.8% of users.

NMH-CS provides several advantages over the original WRF-Hydro/Noah-MP. Unlike the original version that requires compiling for each computer and primarily depends on Unix-like systems, NMH-CS can seamlessly run on Windows systems that support the Microsoft Dotnet Framework. The executable files, once compiled, can be easily packaged and distributed to other Windows machines, offers greater convenience for users who are less familiar with Unix-like operations. The utilization of the C# language facilitates advanced software tools for visualizing and analysing the model's code, enhancing the convenience for users to read, modify and debug the code. This is appealing to the model developers who are proficient in C# language and the object-oriented programming. The design of NMH-CS aligns with the input datasets and configurations specified in the 'namelist' file, ensuring high compatibility with WRF-Hydro 3.0. Leveraging the parallel computing capabilities of C#, both the translated Noah-MP LSM simulation and the river routing simulation within NMH-CS support parallel execution on common personal computers.

## 2 The Noah-MP LSM

Noah-MP is a robust model renowned for its capability to represent diverse physical processes. Since its initial introduction (Niu et al., 2011) (Yang et al., 2011), Noah-MP has been widely used. For example, Noah-MP has been coupled to WRF-Hydro as a major module, and can be seamlessly integrated into the Weather Research and Forecasting (WRF) model (Gochis, 2020). Furthermore, the offline WRF-Hydro model plays a pivotal role in the National Water Model, contributing to the simulation of floods and river flows across the United States (Bales, 2019) (Francesca et al., 2020) (Karki et al., 2021). Noah-MP's versatility extends to applications such as streamflow prediction (Lin et al., 2018) and the estimation of evapotranspiration, surface temperature, carbon fluxes, heat fluxes, and soil moisture, as demonstrated in many studies (Chang et al., 2020; Gao et al., 2015; Li et al., 2022; Ma et al., 2017; Yang et al., 2021). Noah-MP is supported by several different modelling frameworks to facilitate coupling it to various earth system framework models including HRLDAS (Chen et al., 2007), LIS(Kumar et al., 2006), and WRF-Hydro(Gochis, 2020). This makes Noah-MP a powerful research and forecasting tool within the hydrology community.

Noah-MP excels in physical representation of water and energy dynamics across various environmental layers, including a vegetation canopy layer, multiple snow and soil layers, and an optional unconfined

aquifer layer for groundwater. To capture specific physical processes, Noah-MP employs multiple
parameterization schemes, providing users the flexibility to choose from a total of 12 parametrizations,
as detailed in Table 1. This versatility enables tailored representation of diverse environmental conditions
and processes, enhancing the model's adaptability and applicability.

**Table 1:** Parameterization options for Noah-MP (an asterisk (*) denotes the recommended default
option). Certain abbreviations correspond to terms used for Parameterization Schemes (PS) in Noah-
MP, and their meanings can be referenced in the WRF-Hydro 3.0 user document (Gochis et. al., 2015).
Note that these options may not be applicable to other versions of Noah-MP, such as that used in
HRLDAS. The scheme options here presented by some abbreviation marks such as 'Noah' or
Schaake96 are those used in the 'namelist' file for Noah-MP.

| Abbreviation | Physical parameterization | Scheme Code | Scheme Options |
|---|---|---|---|
| DVEG | Vegetation option | 1~5 | *1 table LAI, read FVEG; 2 dynamic LAI, FVEG=f(LAI); 3 table LAI, FVEG=f(LAI); 4 table LAI, FVEG=maximum; 5 dynamic LAI, FVEG =maximum |
| CRS | Stomatal conductance (controls transpiration from leaves) | 1~2 | *1 Ball-Berry; 2 Jarvis |
| BTR | β-factor (soil moisture stress factor controlling transpiration) | 1~3 | *1 Noah; 2 CLM; 3 SSiB |
| RUN | Runoff (runoff generation at and below the surface) | 1~4 | 1 SIMGM; 2 SIMTOP; *3 Schaake96; 4 BATS |
| SFC | Surface layer drag coefficient | 1~2 | *1 M-O; 2 Chen97 |
| FRZ | Frozen soil permeability | Fixed to 2 | *1 NY06; 2 Koren99 |
| INF | Supercooled liquid water | Fixed to 2 | *1 NY06; 2 Koren99 |
| RAD | Radiation transfer option | 1~3 | 1 gap=F(3D,cosz); 2 gap=0; *3 gap=1-Fveg |
| ALB | Snow surface albedo | Fixed to 2 | 1 BATS; *2 CLASS |
| SNF | Precipitation partition option (rainfall or snowfall) | Fixed to 2 | *1 Jordan91; 2 BATS; 3 Noah |
| TBOT | Lower boundary of soil temperature | 1~2 | *1 zero-flux; 2 Noah |
| STC | Snow/soil temperature time scheme | Fixed to 1 | *1 semi-implicit; 2 fully implicit; 3 Ts=f(fsno) |


## 3 Development of NMH-CS

### 3.1 Translation of Noah-MP Code

Our primary focus in developing NMH-CS involves translating the original FORTRAN code of Noah-
MP into the C# language. It is essential to note that this translation is based on a relatively older version
of Noah-MP utilized in WRF-Hydro 3.0, as the process was started before the release of Noah-MP 5.0
(He et al., 2023).
Converting FORTRAN code into C# is not straightforward due to significant syntax differences
between the two languages. The reconstruction of the model in C# follows an object-oriented design.
While FORTRAN is traditionally a function-based language, the core Noah-MP module's functions,
subroutines, and state variables are encapsulated as members within a C# class named GridCell (**Fig.
1(a)**). This class represents all Noah-MP behaviors within a grid box. The variable names, function
definitions, data structures, and execution logic have been kept largely consistent with the original
FORTRAN code, ensuring user-friendliness for those familiar with Noah-MP. To handle the execution
on multiple grid boxes, another C# class named Driver is employed. This class manages tasks such as
initializing model variables, creating multiple grid boxes, reading/writing files, and controlling the
execution of the model.
Throughout the translation process, a key focus was addressing operations on FORTRAN arrays (**Fig.
1(b)**), crucial for representing the state of soil and snow layers in Noah-MP. Unlike C#, FORTRAN
allows arrays to have user-specified index ranges (e.g., index values from -3 to 4). However, in C#, the
first index of all arrays invariably starts from 0. To streamline the translation, we designed a wrapping
class of C# arrays, named FortArray, to mimic FORTRAN arrays. The wrapped inner array data in
FortArray adheres to standard C# conventions, accepting 0 as the inner index of the first element. Yet,
externally, the class allow access to the array values through extra indices by providing methods for
index translation from outer indices (FORTRAN style) to inner indices (C# style):
$$I_{in} = I_{ex} - I_{start} \qquad (1)$$
Where $I_{in}$, $I_{ex}$ and $I_{start}$ represent the inner index, the outer index and the first outer index. The inner index
corresponds to the standard C# arrays, while the outer index corresponds to the FORTRAN arrays. For
instance, if a FORTRAN array of 8 elements has an index range from -3 to 4, it will be translated into a
FortArray that has a standard inner array of 8 elements, accompanied by two arguments representing the
starting FORTRAN index (-3) and the ending FORTRAN index (4), but the range of its inner indices
remain 0~7. This array translation technique ensures that all the original execution logic in Noah-MP is
seamlessly preserved in NMH-CS.
The model also supports parallel execution, implemented through the native parallel functionality of the
C# language. The technique can efficiently allocate computational tasks over the grid boxes to different
threads which can be executed by separate CPU cores. For instance, if a grid domain requires the
execution over 2400 grid boxes, and the tasks are assigned to 8 threads, each thread is responsible for
the calculations on approximately 300 grid boxes. It's crucial to note that if the number of specified
threads exceeds the number of CPU cores, several threads should be executed by the same CPU core.
Therefore, specifying more threads than the available CPU cores does not contribute to an overall
improvement.

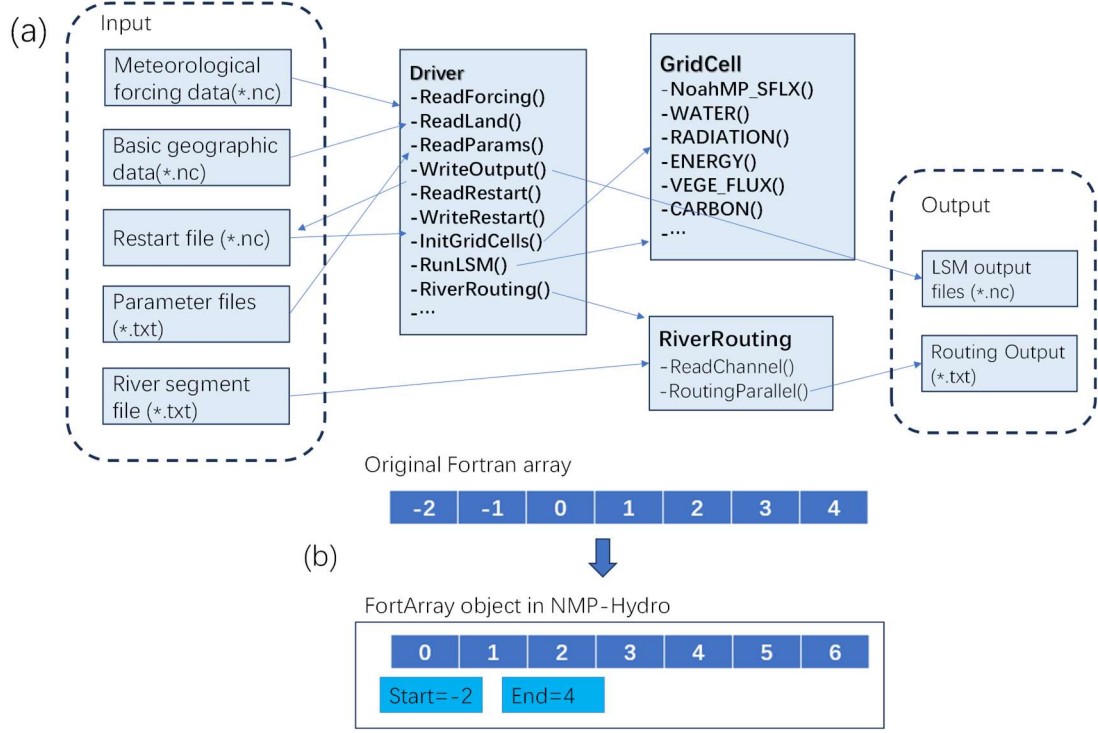

Figure 1. The architectural diagram of NMH-CS (a) and the conversion of FORTRAN arrays to C# arrays (b). NMP-Hydro is a reconstructed replica of the version of Noah-MP that is coupled in WRF-Hydro 3.0.

**3.2 Coupling with a parallel river routing module**

The Noah-MP land surface model can produce column outputs of runoff, but cannot simulate the horizontal movement of water. In order to simulate the surface movement of runoff in the river channels, we integrated a parallel river routing module, which is based on the Muskingum method for vectorized channel networks. The module is described in a previous study (Liu et al., 2023). This module is not the previous RAPID model that was coupled in WRF-Hydro (Lin et al., 2018). This parallel river routing module, implemented using C#, incorporates our unique techniques.

The first technique is an array-based sequential processing method for Muskingum routing. Muskingum-Cunge equation (Cunge, 1969) with lateral inflow considered is

$$Q_{e,t+1} = C_0 * Q_{s,t} + C_1 * Q_{s,t+1} + C_2 * Q_{e,t} + C_3 * Q_{lat,t+1} \qquad (2)$$

Where

$$C_0 = \frac{kx + 0.5\Delta t}{k(1-x) + 0.5\Delta t}, \quad C_1 = \frac{-kx + 0.5\Delta t}{k(1-x) + 0.5\Delta t}, \quad C_2 = 1 - C_0 - C_1, \quad C_3 = \frac{\Delta t}{k(1-x) + 0.5\Delta t}$$

Here, $Q$ represents the channel streamflow (m³/s), which can be considered as a function of time and position; $s$ is the start point of a channel segment/node; $e$ denotes the end point of the channel segment, both of them are used as subscriptions for different spatial positions; $t$ denotes the start of the period or inflow, $t+1$ denotes the end of the period or outflow; Both $t$ and $t+1$ are used as subscriptions for position. The subscription 'lat' represents the lateral streamflow(runoff) from current river catchment.

Two parameters of the Muskingum-Cunge method are $k$ and $x$. $k$ is the travel time of a flow wave with celerity $c_k$ through a channel segment of length $L$, thus $k = L/c_k$. Parameter $x$ can be estimated by

$$x = \frac{1}{2}\left(1 - \frac{q}{S_o c_k L}\right) \qquad (3)$$

Where $q$ represents unit width streamflow, $S_o$ is the channel bed slope.

The lateral inflow of each river segment is the runoff simulated by column Noah-MP LSM in NMH-CS, which is expressed as

$$Q_{\text{lat},t} = R_{c,t} A_s \qquad (4)$$

Where $Q_{\text{lat},t}$ is the lateral inflow at time $t$, $R_c$ is the runoff value at the given grid box, $A_s$ is the local catchment area of current channel segment.

The form of Eq.(2) implies that the river routing calculation can be completed from any upstream river segment to its downstream segment. At the time $t+1$, all the outflows ($Q_{\text{e, }t+1}$) of multiple upstream segments will be summed up as the inflow ($Q_{\text{s, }t+1}$) of current segment. Although a common river network is a tree-like structure, it can be represented as a sequential array, where any upstream river segment is stored near the array head (with zero index), while its downstream river segment is stored near the array's tail (with large array index). Therefore, the Muskingum routing can be calculated over the river segment array in a unidirectional processing way (please see Fig.1 in Liu et al. (2023)).

The second technique is the straightforward equal-sized domain decomposition method to conduct parallel calculation: just allocating the river segments into equal-sized blocks. Within each block, the Muskingum routing can be executed separately by a single CPU core. This treatment is based on that in any block, most segments have upstream segments within the same block, and only a small fraction of the segments have upstream segments in other blocks. Therefore, all river segments that receive inflows from other blocks (referred to as 'cross-block segments') need to be identified. These cross-block segments should be executed by the primary core, after the multi-domain parallel execution is completed.

The third technique is a specific sorting approach for river segments used in domain decomposition. It has been proven that a depth-first traverse of the river segments is more suitable for the parallel execution of the Muskingum method, compared to a width-first traverse, due to less cross-block segments in the blocks.

This module requires two additional inputs files, a river segment list file named 'ChannelOrder.txt' and a 'namelist.txt' file. The latter file is used to set parameters and the length of time step. Each river segment in the list file presents following information: its own index, the index of its next downstream river segment, the row number and the column number of the grid box (in Noah-MP's running domain) providing runoff input to the current segment, the length (m) of the current river segment, the two parameter values ($K$ and $X$) of the Muskingum method, the area of the catchment of the current segment.

The river segment list can be derived from both gridded river network or vectorized river network. The resolution of the river routing is determined by the original river network from which river segments is derived. Therefore, the choice of using vector river network or gridded river network and the selection of spatial resolution are completely determined by the users. The length of the temporal step of the river routing is required to be multiple times shorter than the time step for running the Noah-MP, and can also be designated by the users. For example, the time step of routing is set to 600s, while the time step for Noah-MP LSM is usually set to 3 hours.


This approach's primary advantage lies in its ability to simply decompose any river network into multiple
domains with an equal number of river segments. Achieved by evenly dividing the river segment list into
any number of blocks, this innovation capitalizes on the inherent tree-like structure present in most river
networks. Importantly, it does not necessitate consideration of the topological conditions specific to a
given network, as required in studies such as Mizukami et al. (2021) or David et al. (2015). This design
allows parallel execution of river routing on modern personal computers equipped with multi-CPU cores.

The integration of the river routing module with the Noah-MP LSM involves assigning lateral inflows
from the LSM-simulated total runoff to the river routing model. In the present NMH-CS configuration,
we utilize a catchment centroid-based coupling interface (David et al., 2015). This method designates
the LSM grid cell containing the catchment centroid (referred to as the "centroid cell") as the location
for a river reach to receive lateral inflows. At a specific temporal step, the computed contributing runoff
discharge $Q_{lat}$ (unit: m$^3$/s) is determined by the following expression:
$$Q_{lat} = R(nx, ny) \times F \times 1000 \qquad (5)$$
where $R(nx, ny)$ is the runoff (mm, surface + subsurface) simulated by the LSM during the time step,
F is the catchment area (km$^2$) contributing water to the current river segment.

Alternatively, employing weighted assignments from different grid boxes, akin to the method utilized in
(Lin et al., 2018), is also a valid approach. However, this method requires the generation of weights from
multiple grid boxes. Given the rough resolution of the meteorological datasets, each grid box can
encompass the catchment areas of multiple river segments, the coupling approach using area weighting
is unlikely to yield substantial improvements for most river segments.
**3.3 Code Debugging Process**
To eliminate any potential code errors resulting from incorrect translation, we conducted a thorough
checking of the code by performing model execution benchmark tests on single-column running on
specific grid boxes. Here, In the large domain (the same domain described in section 4.1), each time the
grid box for single-column execution is arbitrarily selected. Such debugging tests were carried out in two
approaches. The first approach was carrying out a meticulous step-by-step debugging by examining the
printed values of many variables (including many local variables in the code) in WRF-Hydro 3.0. This
process was also repeated by switching each option of multiple physical parameterization schemes. The
grid box for the single-column debugging was also switched several times. Such debugging has been
conducted numerous times and has effectively eliminated any code errors arising from inaccurate
translation. Although meteorological driving data for the debugging simulation is prepared for the period
between 2000 and 2016, such debugging tests are only feasible for a limited number of temporal steps
for a grid-box execution. It is also impossible to conduct debugging on the entire domain.
The second approach is an artificial code checking process. Considering that the stepwise debugging
through years-long simulations is impractical, we checked the NMH-CS's code by comparing with the
original FORTRAN code for many times. Through this checking, many code inconsistences were
identified and corrected.

# 4. Testing of NMH-CS

## 4.1 Application area and data

### 4.1.1 Application area

The Yellow River Basin in Northern China is used as a test area of NMH-CS. The gridded domain, as illustrated in **Fig.2-3**, encompasses the entirety of the Yellow River Basin (YRB) and most of North China, comprising of 350 columns and 170 rows, with a resolution 6 km in Lambert conformal conic projection coordinates. Geophysical data essential for the domain, including digital elevation, land use and land cover, and green vegetation fraction, were extracted from the WRF/WPS 3.5 input database.

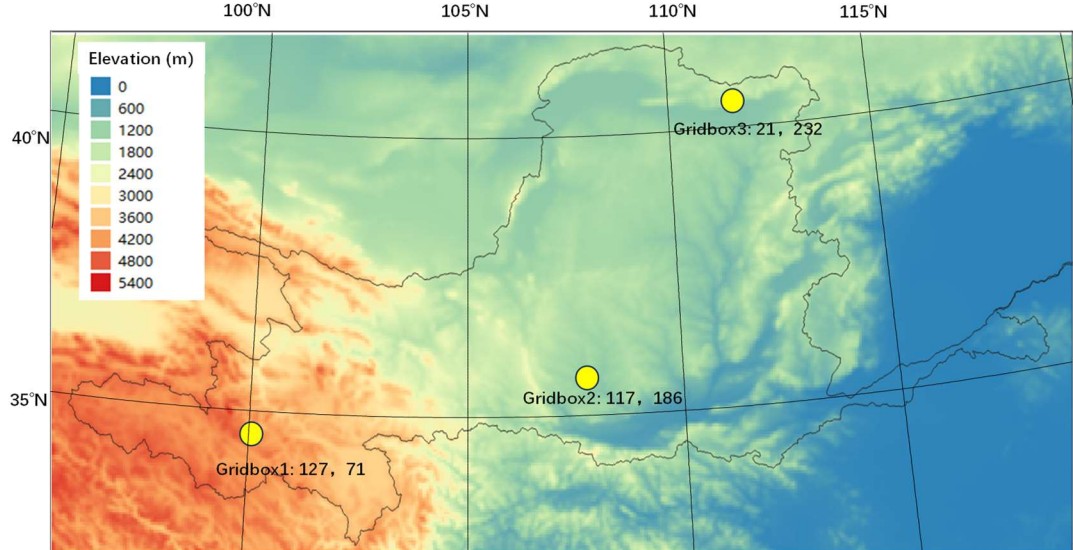

**Figure 2.** The terrain map of the simulation domain and the three grid boxes for comparison of state variable time series. The three grid boxes for extracting state variables to compare are represented by yellow dots, marked with grid box code, the row number and the column rows.

For the river routing simulation, the digital river network of the Yellow River was obtained from the HydroSHEDS dataset (version 1) (http://hydrosheds.cr.usgs.gov/). HydroSHEDS was derived from gridded digital elevation data with a resolution of 15 arc-seconds. Given the substantial human intervention and the intricate nature of reproducing observed daily or hourly water discharge, uniform values were assigned to all river segments for the river routing parameters (specifically, the wave celerity ($c_k$) and another parameter ($x$) describing the river channel condition, as detailed in David et al. (2013). No precise calibration is required here, as monthly or annual river discharge remains unaffected by changes in routing parameters.

The Yellow River basin experiences significant human impacts, including irrigation, industrial water usage, and groundwater extraction. Major artificial reservoirs and numerous smaller reservoirs regulate the river's discharge, serving as the primary water resource during the dry season. However, such extensive human interference presents substantial challenges in accurately modeling river discharge. Comparatively, the river discharge upstream of the Lanzhou (In Zone 1 as shown Fig.3) hydrological station contributes over half of the entire YRB's total discharge, and is relatively less impacted by dams, enabling us to test the model's performance.

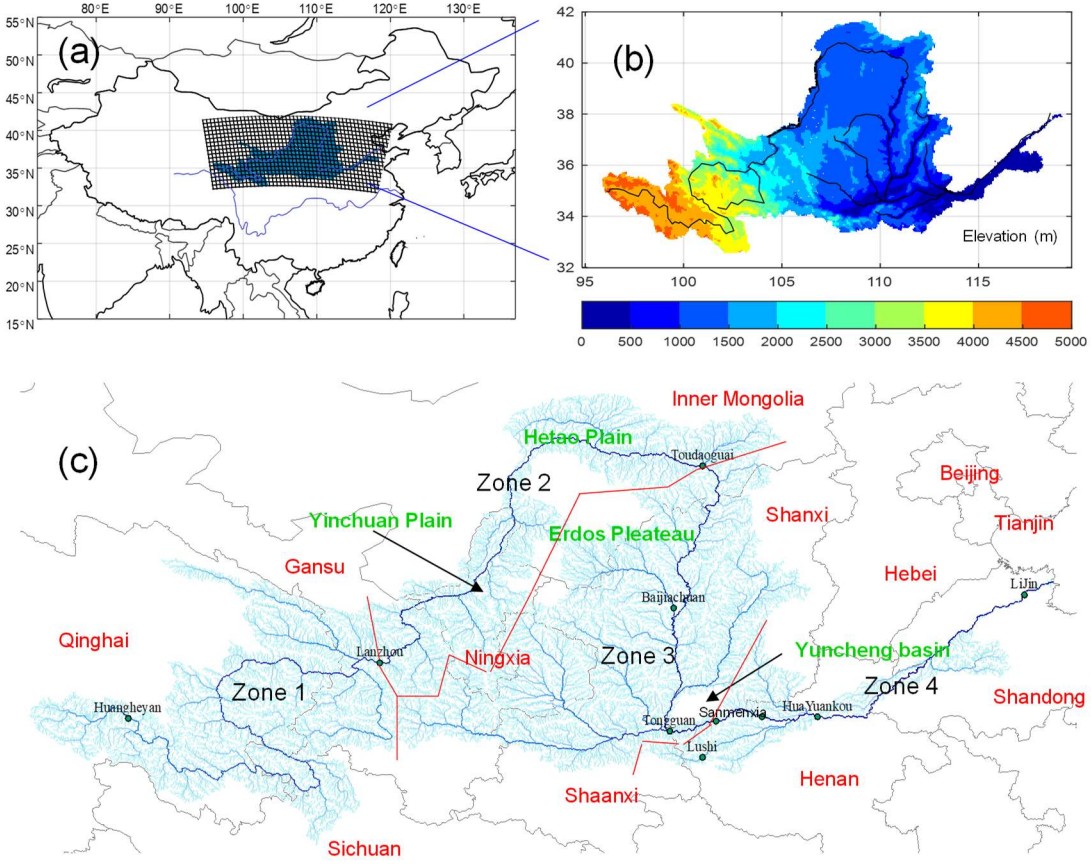

Figure 3. The grid domain covering the Yellow River Basin and the North China area: (a) Geographical location within China; (b) Elevations; (c) Vector River networks utilized for river routing modeling (extracted from the HydroSHEDS dataset). The delineation of the boundaries between distinct zones controlled by four gauging stations (Lanzhou, Toudaoguai, Sanmenxia, Lijin) is represented by red lines.

**4.1.2 Meteorological dataset and river discharge data**

To drive the two models (NMH-CS and WRF-Hydro), the same 3-hourly and 6-km grided meteorological forcing dataset comprised of shortwave and longwave downward radiation, wind velocity, air temperature, relative humidity, air pressure at the surface, and precipitation rate was acquired. The benchmark dataset was clipped and regrided (bilinear interpolation) from the $1.0° \times 1.0°$ GLDAS-1 land surface product (Gan et al., 2019; Rodell et al., 2004), for the period 2000-2016. Given the limited availability of observational river discharge data between 2001 and 2016, the extracted data pertains to this period, and additional data between 1996 and 2000 was also extracted for the model's spinning-up. In previous research, the spinning-up of Noah-MP requires 50 years (Wu et al., 2021) or more than one hundred years (Zheng et al., 2019) to achieve an equilibrium state. However, in this study, the spin-up process was conducted in two steps. In the first step, the period from 1996 to 2016 was run three times to generate a 'restart file' for a 63-year spinning-up, utilizing the initial PS combination. In the second step, starting from this initial combination, new schemes were adopted, and the 'restart file' obtained from the initial scheme combination was used to initiate the formal experiments covering the period from 1996 to 2016.

The dataset of Natural River Discharge (RND) reconstructed by the Yellow River Conservancy

Commission of the Ministry of Water Resources was gathered to assess the model output. Annual natural
discharges from the monitoring station of Lanzhou were collected for the period from 2001 to 2016.

**4.2 Running speed**

Compared to other differences between the two models, running speed is the least important factor to
consider. Considering that NMH-CS and WRF-Hydro run on different platforms (Windows or Linux)
and machines, it is also difficult to achieve a comparative evaluation of running speed. Actually, the
comparison of running speed depends on the programming language used. In theory, FORTRAN and C
programs can run faster than C# programs because FORTRAN and C are relatively low-level languages
compared to any modern object-oriented language. However, as a language that can run in native
machine code, C# is not slow. This means that there won't be a large difference in running speed between
C# and FORTRAN. There are few authoritative publications on the running speed of these two languages,
but there have been many documents on benchmark testing on the internet. When considering parallel
execution, comparing the running speed of the two models also become more unnecessary. NMH-CS can
run in parallel mode on personal computers, while WRF-Hydro does not have this functionality. On the
other hand, WRF-Hydro can run in parallel mode in the Message Passing Interface (MPI) environment
of high-performance computers, while NMH-CS does not support MPI.
We tested the execution time of NMH-CS by setting different numbers of C# parallel threads. The
computer used for the testing is a common laptop with 6 CPU cores. The results indicate that for the
execution of the entire domain, as the number of threads increases from 1 to 6, the average time consumed
per time step is 1576ms, 977ms, 801ms, 711ms, 679ms, and 672ms, respectively. When the number of
threads is set to 1, the time spent is slightly greater than the execution time in the non-parallel mode
(1461ms). It is worth noting that the time spent is not linearly related to the number of parallel threads,
which can be explained by various reasons. One is that some tasks are not actually executed in parallel
mode, such as reading meteorological input files. Another reason is that not all threads in NMH-CS are
fully processed by the CPU cores, as there are many other tasks in the entire Windows environment that
have to be processed simultaneously by the CPU cores.

**4.3 Comparing the outputs of NMH-CS and WRF-Hydro**

It is noteworthy that there are numerous parameterization scheme combinations for Noah-MP, which
makes it unfeasible to compare the results generated under all scheme combinations. Therefore, the
output of NMH-CS and WRF-Hydro was compared only with the default parameterization scheme
combination, based on the exact same meteorological dataset. The comparison was conducted in two
ways. The first comparison is that of the spatial maps of multiple variables (Table 2) for a specific year
or day. For each state variable, such as SFCRNOFF, the maps of state variables simulated by NMH-CS
and WRF-Hydro are presented. The difference ($\Delta$) between the values of the same variables simulated
by the two 'Noah-MP models' is calculated as
$\Delta = V_{\text{NMH-CS}} - V_{\text{WRF-Hydro}}$         (6)
Where $V_{\text{NMH-CS}}$ and $V_{\text{WRF-Hydro}}$ are the state variable simulated by the two Noah-MP models, respectively.
For certain variables, such as SFCRNOFF, UGDRNOFF, ECAN and ETRAN, the percent relative
differences were also calculated as follows:
$\delta = 100 \cdot \Delta / V_{\text{WRF-Hydro}}$         (7)
The second method is to compare the temporal variations of state variables at specific grid boxes. In this

case, only three grid boxes (Fig. 2) were selected to extract the state variable time series. The selection of these grid points is an arbitrary decision made by roughly considering different climate zones, without other strict consideration. Gridbox1 is selected from the Qinghai Plateau region, corresponding to the source region of the Yellow River. Gridbox2 corresponds to a location of Inner Mongolia and the north of Loess Plateau. Gridbox3 is in a hilly area of the Wei River Basin (a major part of the Yellow River Basin). In fact, during this study, other grid points also have been casually tested, but the results are mostly similar to the above mentioned 3 grid boxes. and will not be presented in the paper.

**Table 2** The state variables simulated by NMH-CS and WRF-Hydro, which is verified by generating maps

| Variable name | description | unit |
|---|---|---|
| SFCRNOFF | Accumulated Surface runoff | mm |
| UGDRNOFF | Accumulated ground runoff | mm |
| ECAN | Evaporation from canopy | mm |
| ETRAN | Vegetation transpiration | mm |
| TV | Vegetation temperature | K |
| TG | Ground temperature | K |
| SOILT | The temperature for soil layers | K |
| SOILW (or SH2O) | The volumetric content of moisture in soil layers | $m^3 \cdot m^{-3}$ |
| SNOWH | The total depth of snow layer | m |
| SNEQV | Snow water equivalent | $kg \cdot m^{-2}$ |
| EAH | Canopy air vapor pressure | Pa |
| EVG | Ground evaporation heat | $W \cdot m^{-2}$ |
| CHV | Exchange coefficient vegetated | $m \cdot s^{-1}$ |
| CHLEAF | leaf exchange coefficient | -- |
| TR | Transpiration heat | $W \cdot m^{-2}$ |
| EVB | Evaporation heat to atmosphere bare | $W \cdot m^{-2}$ |
| FIRA | Total net long-wave radiation to atmosphere | $W \cdot m^{-2}$ |
| TRAD | Surface radiative temperature | K |
| ALBEDO | Surface albedo | -- |

**4.3.1 Maps of state variables**

To test whether NMH-CS can produce the corresponding outputs of the original WRF-Hydro (Fortran-version Noah-MP), many state variables (Table 2) from multiple time slices have been checked by drawing maps. Only four slices (10 June 2000, 10 June 2001, 10 June 2004, and 10 June 2008) were arbitrarily selected here without special consideration. Only some maps of these state variables at certain time slices are presented in both the paper and the supplementary information. The maps for all the sate variables in Table 2 reflect high consistence between NMH-CS and WRF-Hydro, with only the maps for four representative variables (SFCRNOFF, UGDRNOFF, TV and TG) are shown in **Fig.4 and Fig.5**. As can be seen, there is visually little difference in the spatial patterns of the results. Similarly, no discernable visual difference is also apparent for the maps of other variables. However, the relative difference of annual surface runoff and annual underground runoff is significantly large at some areas

(generally in high-elevation regions), where NMH-CS underestimated/overestimated those values above 10% (**Fig.4**). These regions with large relative differences of underground runoff actually are in small absolute differences, primarily because the annual total groundwater runoff in these areas is inherently low (<50 mm). This discrepancy is likely attributable to floating-point arithmetic errors, but the possibility of other contributing factors cannot be ruled out.

For most of the domain, the difference in TV is smaller than 0.2 ℃, but in some sporadically districted locations, the TV's difference can be larger than 2 ℃ (**Fig.5**). The comparison of TG has the similar effects, but the difference is more significant than that of TV. The similar high consistency effects are also reflected by other state variables, including soil temperature, soil water content, snow water equivalent (**Fig.S2-S4** in supplementary information). The differences in these state variables between the two models are generally small, except some large ones sporadically distributed in the high-latitude areas.

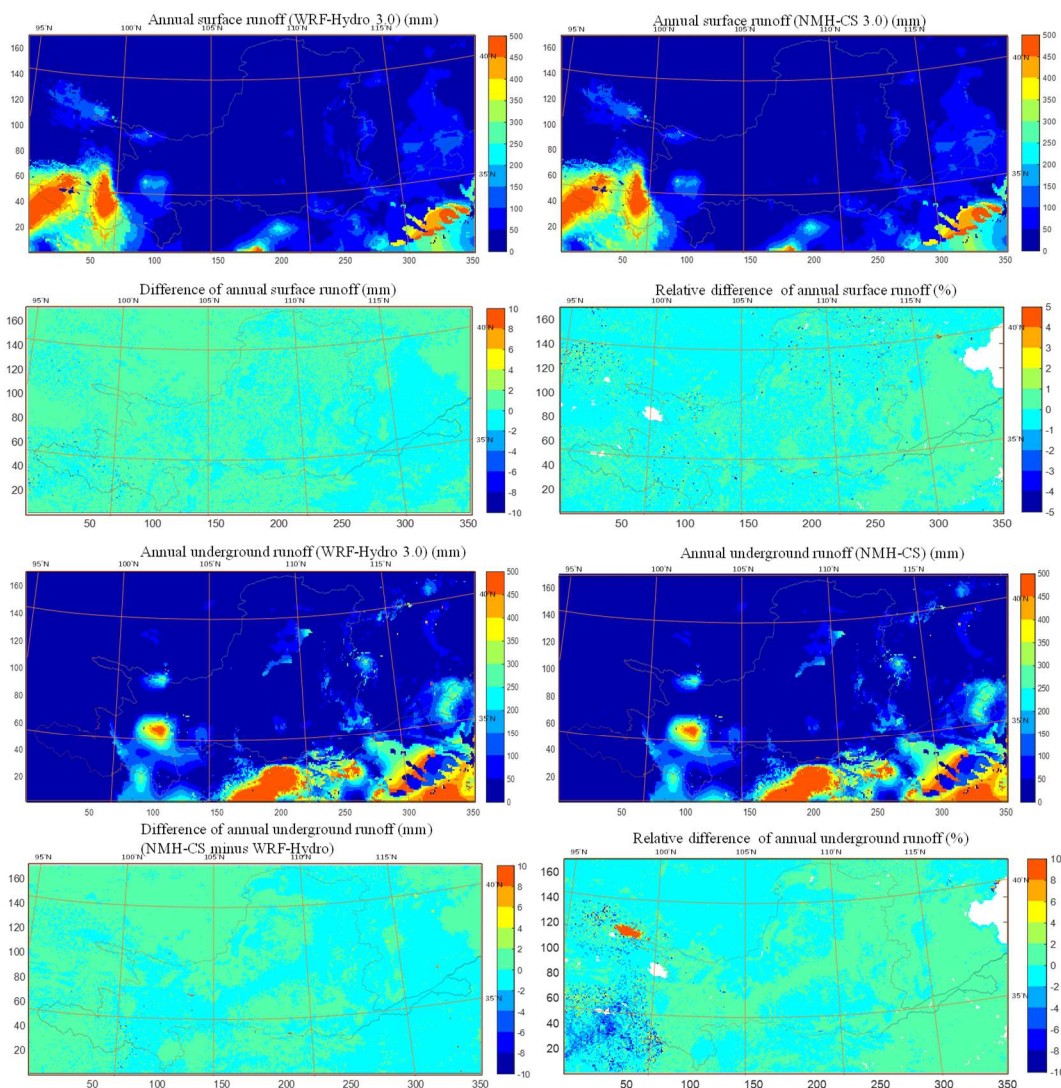

**Figure 4.** Maps of annual total values, differences, and relative differences of SFCRNOFF (surface runoff, mm) and UGDRNOFF (underground runoff, mm) simulated by WRF-Hydro3.0 and NMH-CS 3.0, for the year 2005. The labels for horizontal axis and vertical axis are row numbers and column numbers of the grid domain respectively.

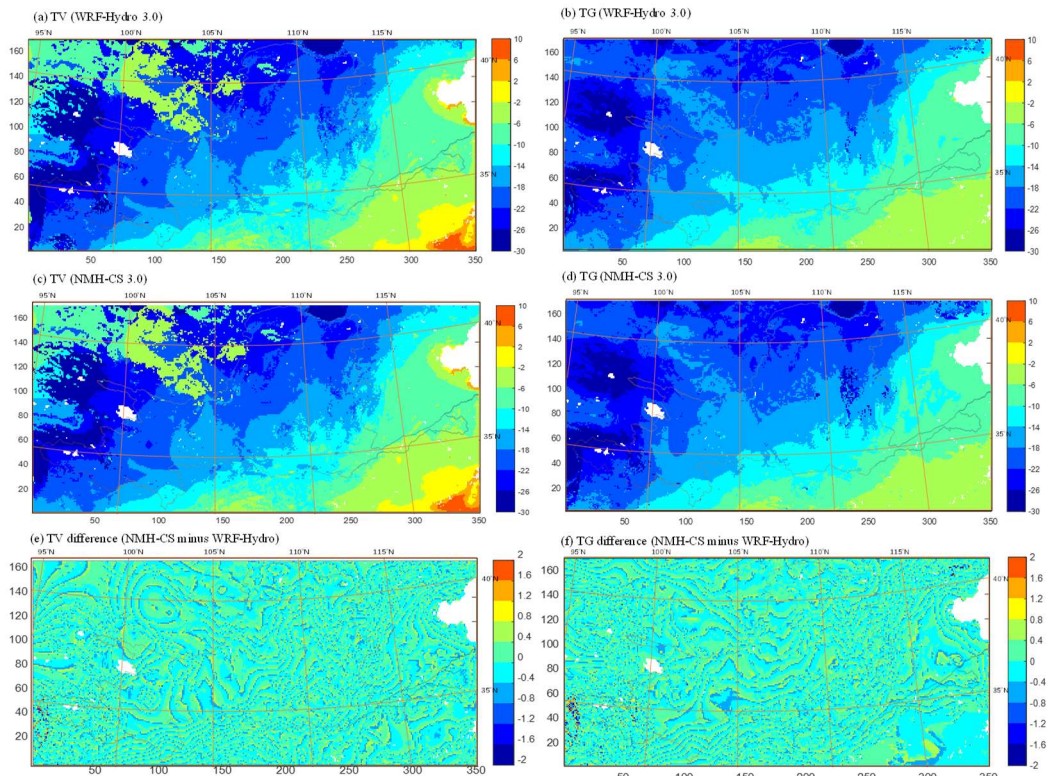

**Figure 5:** Similar to Fig.4, but the maps for TV (vegetation temperature, ℃) and TG (ground temperature, ℃), for the day Jan. 1st, 2008.

**4.3.2 Temporal variations of state variables**

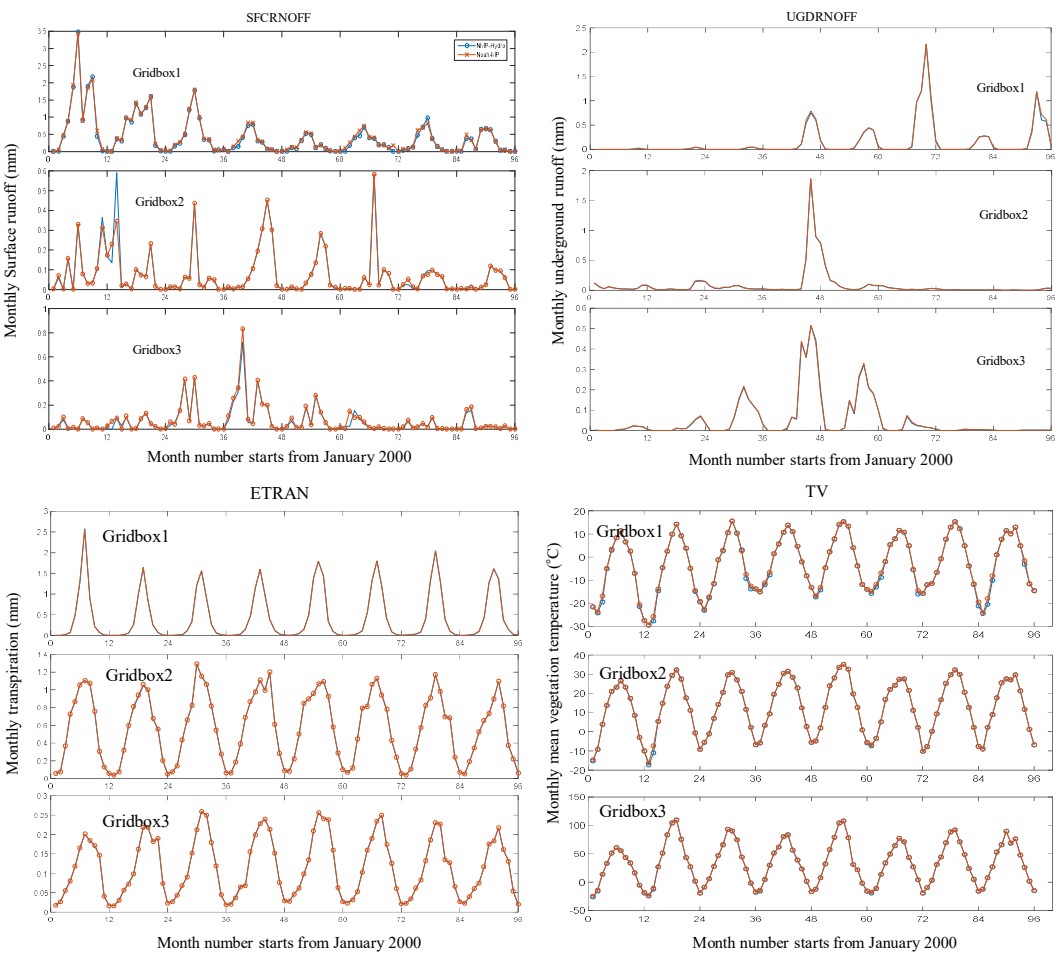


**Figure 6:** Monthly surface runoff (SFCRNOFF in mm), underground runoff(UGDRNOFF in mm),
transpiration (ETRAN in mm) and vegetation temperature (TV in ℃) simulated by WRF-
Hydro3.0(blue) and NMH-CS (red) at the three grid boxes, 2000-2007.

The outputs at the three representative grid boxes (as shown in **Fig.2**) indicate that the two models
produced consistent temporal changes (**Fig.6, Fig.S5 and Fig.S6**). For certain variables, for example,
SFCRNOFF, TV and TG (other variables as well), occasionally, some significant differences were
found in certain months for Gridbox2. It must be such occasionally happened differences that caused
the spatial disparity as shown in **Fig.4-5**. We checked the disparity at certain grid boxes on the 3-hourly
values and found that the differences also happen sporadically (**Fig.7**). Almost all the disparities occur
during the cold months (November, December, January, and February). However, it is worth noting that
mostly the simulated state variables in these months show no difference. Considering such mismatch
usually happens in cold months and high-elevation regions, it may be caused by the different
calculation accuracy for the processes of snow or frozen soil. For the three representative grid boxes,
no significant differences were identified for certain variables, including many variables like TR, EAH,
TV, ETRAN, UGDRNOFF (no plots for these variables will be presented in the paper).
By comparing and analyzing the printed state variables (in 3-hourly timesteps) in WRF-Hydro and the
NMH-CS, we found the major inconsistencies occur in the module of snow water (named
'SNOWWATER' in the code). From **Fig.7**, three major inconsistencies simulated between the two
models occur usually simultaneously in the multiple state variables. These three cases demonstrates
that almost all the major inconsistencies in multiple variables are caused by the minor inconsistencies
in SNOWH (the state variable to indicate the depth of snow). The logic of snow process in Noah-MP is
coded as when SNOWH is below 0.025m, the ISNOW (a state variable to indicate whether snow a
layer exists) is set to zero (no snow layer), otherwise, is set to 1 (having a snow layer). Therefore, if
SNOWH simulated by NMH-CS is close to 0.025, a small floating-point error may trigger a division
between having a snow layer and no snow. Due to the different physical effects of radiation balance
between snow layers and the ground, the distinction between having a snow layer and no snow layer
will further lead to significant inconsistencies in snow depth (SNOWH), snow water equivalent
(SNEQV), soil water (SOILW), vegetation temperature (TV), and ground temperature (TG). Once an
inconsistency occurs, it will persist for a period of time. It is highly probable that the minor differences
in SNOWH is caused by accumulation of floating-point error, because for most of the times the
differences are very small except those during the inconsistent periods. This explanation may account
for the inconsistencies observed in the time series in Fig. 7 and the sporadically distributed
discrepancies in high-latitude regions depicted in Fig. 4.

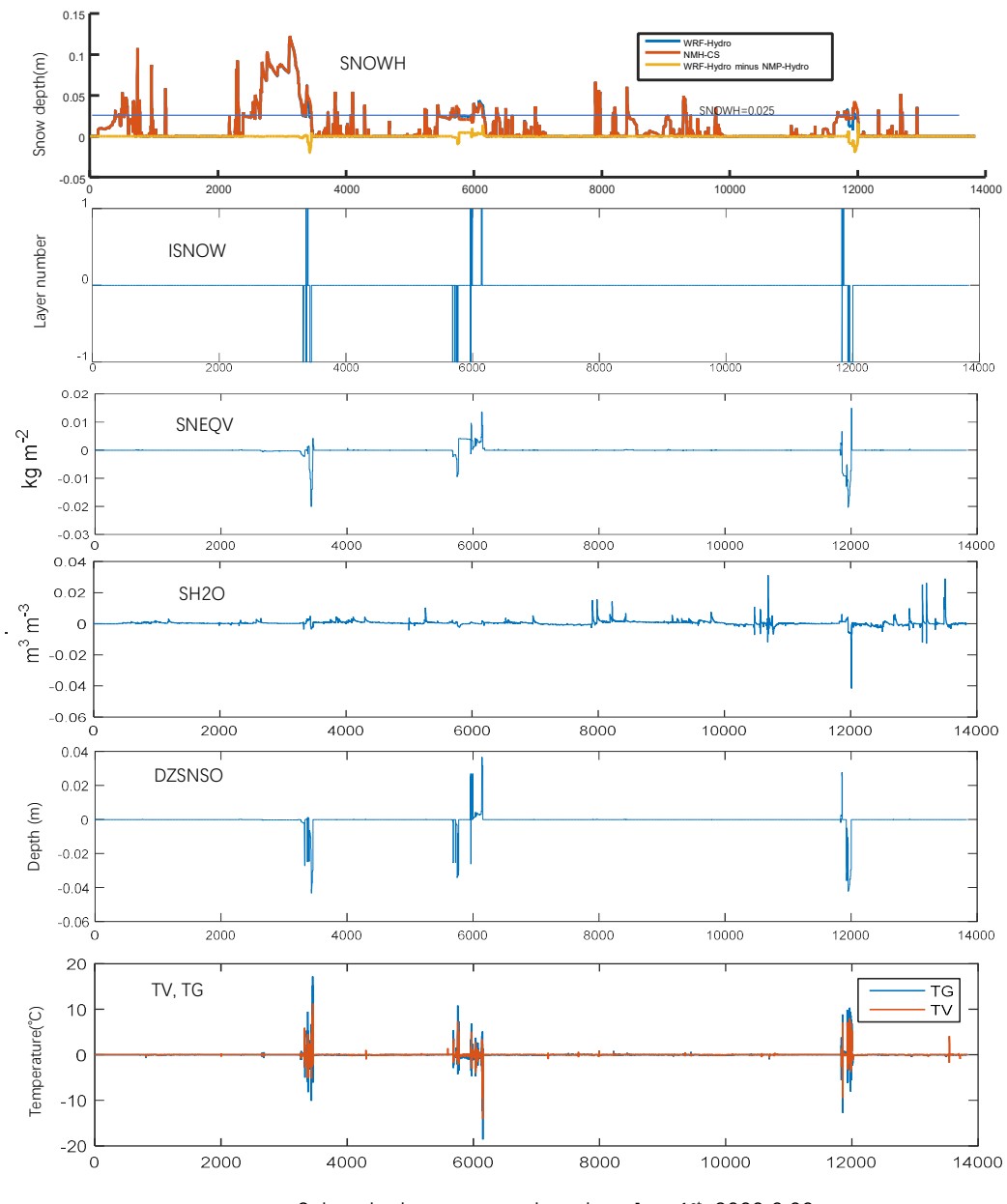

3-hourly timestep number since Jan. 1st, 2000 0:00

**Figure 7:** The differences (NMH-CS minus WRF-Hydro) between the three-hourly variables simulated by NMH-CS and those simulated by WRF-Hydro. SNOWH: snow depth (m); ISNOW: Number of snow layers, count; SNEQV: Snow water equivalent (kg·m⁻²); SH2O: soil liquid water content (m³·m⁻³), equivalent to SOILW; DZSNSO: snow/soil layer depth (m). These variable names are those used in the programming code. The occurring of the three inconsistencies correspond to the short periods: March 4, 2001, January 15, 2002 to February 8, 2002, and January 22, 2004 to February 9, 2004.

The daily time series from multiple grid boxes (including the three in Fig.2) were extracted and compared between NMH-CS and WRF-Hydro. Similar effects were obtained for the grid boxes, but only the results for Gridbox3 are presented as representative in **Fig.8**. It is evident that EDIR, SFCRNOFF, soil water content (SOILW) and TV exhibit small discrepancies, whereas TG demonstrate large disparities. The daily samples for soil temperature, soil water content, snow depth and snow water equivalent are presented in **Fig.S1, Fig.S5 and Fig.S6** (supplementary material). The comparisons about soil layers

reflect that the soil temperature has relatively large inconsistencies, which should also be explained by
the different division of snow layers that is caused by error when SNOWH approaches 0.025m.

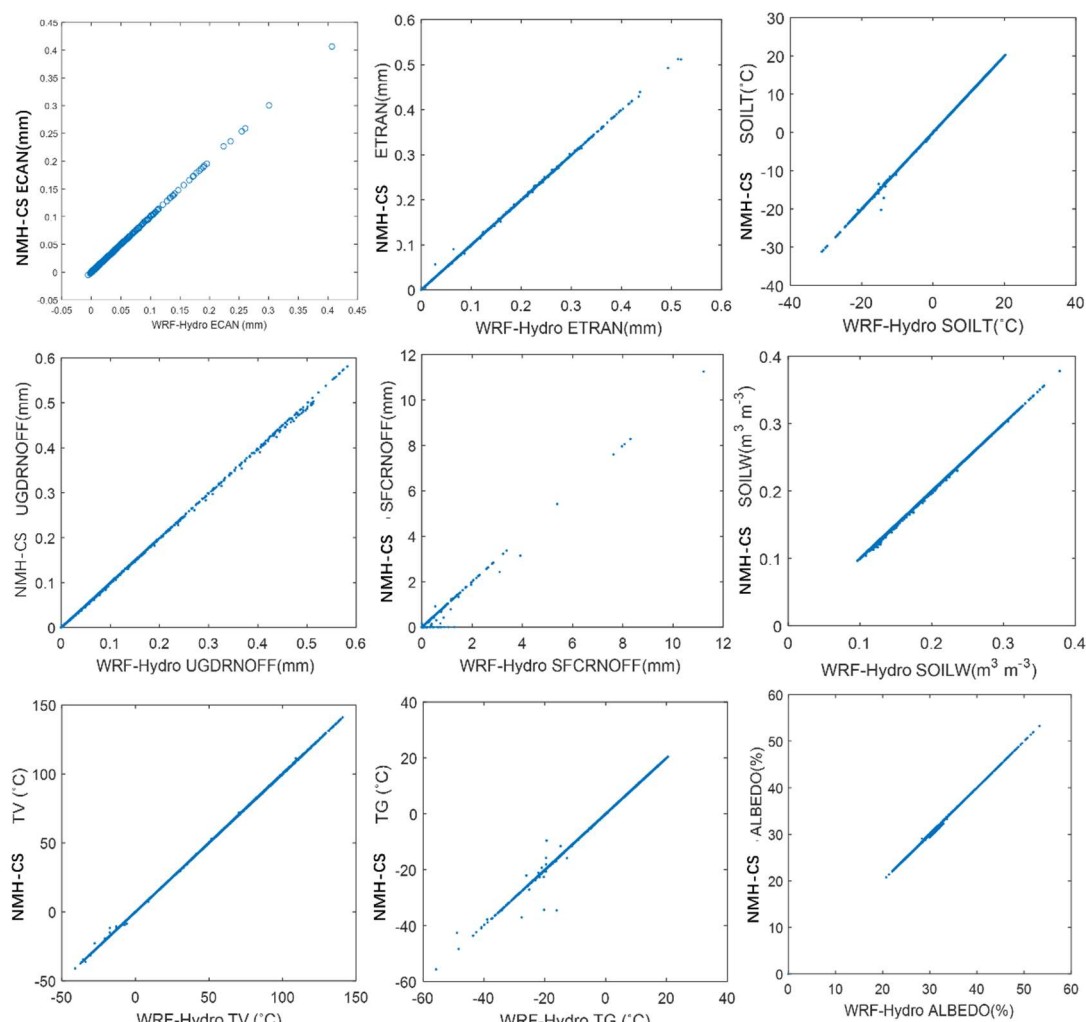

**Figure 8.** Daily state variables simulated by NMH-CS versus by WRF-Hydro at Gridbox3. Due to the
high consistency for most of the values, statistical evaluation metrics such as correlation coefficients or
relative biases will not be presented in the paper.

**4.4 Streamflow discharges for the Yellow River by NMH-CS**
**4.4.1 Experimental design of Noah-MP simulation**
Here, we present the numerical outputs of NMH-CS on the streamflow discharges over the Yellow
River, with various parameterization schemes used. To verify whether the various parameterization
schemes (PSs) of NMH-CS can produce reasonable discharge for the Yellow River catchment area,
this study conducted 17 Noah-MP simulations using different PS combinations. Given the challenge
of determining the relative importance of each parameterization and the impracticality of including
all possible combinations, we adopted a strategic approach. A fixed PS combination was established
as a foundation, and alterations were made to one parameterization's scheme at a time (refer to **Table**
**3**).
In addition to our selected parameterizations, we considered commonly used PS combinations,
including the 'default' combination proposed by Noah-MP developers. Sensitivity analysis was
conducted by analyzing the differences or variations among these incomplete PS combinations. It
is important to note that the chosen PS combinations represent only a subset of all possible
combinations, and the assumed sensitivities based on this subset are considered indicative of overall
sensitivities based on the complete set of combinations.
The PS combinations are represented by codes consisting of sequential digital numbers. For instance,
the default combination is denoted as '11131-1132-111', where each number signifies a scheme
option. The initial experiment, arbitrarily set as the PS combination of '11131-2222-121', served as
the foundation for subsequent experiments. Fifteen experiments (refer to Table 2) were then
conducted by modifying one option at a time from the initial experiment.
These experiments are categorized into multiple groups, with the initial experiment '11131-2222-
121' being employed in multiple groups:
Runoff scheme group (four experiments, switching between: 1. SIMGM, 2. SIMTOP, 3. Schaake96,
4. BATS);
Vegetation scheme group (five experiments, switching between the first option and the fifth option,
see Table 1);
β-factor option group (three experiments, switching between Noah, CLM, and SSiB);
Radiation transfer option group (three experiments, switching between three options);
Group for the scheme of the lower boundary of soil temperature (six experiments);
Group for stomatal conductance scheme (two experiments, switching between two options).
**Table 3.** Experiments conducted in this study

| Number | PS combination code | Abbreviated code | Description |
| --- | --- | --- | --- |
| 1 | 11131-2222-121 | 11131 or 11131-222 or 11131*121 | The control experiment |
| 2 | 11111-2222-121 | 11111 | Experiments with RUN |
| 3 | 11121-2222-121 | 11121 | |
| 4 | 11141-2222-121 | 11141 | |
| 5 | 21131-2222-121 | 21131 or 21131*121 | Experiments with DVEG |
| 6 | 31131-2222-121 | 31131 | |
| 7 | 41131-2222-121 | 41131 | |
| 8 | 51131-2222-121 | 51131 or 51131*121 | |
| 9 | 11231-2222-121 | 11231 | Experiments with BTR |
| 10 | 11331-2222-121 | 11331 | |
| 11 | 11131-2212-121 | 11131-221 | Experiments with RAD |
| 12 | 11131-2232-121 | 11131-223 | |
| 13 | 12131-2222-121 | 12131 | Experiments with CRS |
| 14 | 11131-1132-111 | 'default' | The default PS combination proposed by Noah-MP |

| | | | |
|---|---|---|---|
| | | | developers |
| 15 | 11131-2222-111 | 11131*111 | Experiments with TBOT |
| 16 | 21131-2222-111 | 21131*111 | |
| 17 | 51131-2222-111 | 51131*111 | |


### 4.4.2 Simulated streamflow under various parameterization schemes

The Taylor diagrams (Taylor, 2001) are used to evaluate the different PS on the river discharge at the Lanzhou station. Taylor diagram provides a graphical representation of a model's simulation performance, encompassing three key indices: correlation coefficient ($R$), root-mean-square error (RMSE), and standard deviation (SD).

The streamflow discharges were produced by coupling the NMH-CS with the parallel river routing model. A preliminary comparison on the various scheme combinations is presented in **Table 4**. The monthly performance is summarized in Table 4 based on the comparison of the different PSs as shown in **Fig. 9**. It can be observed that for the majority of parameterizations, the discharges in winter are not sensitive to the schemes, this is to be expected, given the minimal runoff during this season. The simulated summer discharges exhibit notable degree of sensitivity with regard to the various parametrization schemes. In relation to the runoff parametrization, the results obtained through the utilization of the SIGGM scheme led to overestimation during the winter season and underestimation during summer, signifying that considering groundwater could enhance the simulation accuracy of the catchment modulation as opposed to other schemes.

For the Lanzhou station, over 50% of experiments produced discharges with correlations larger than 0.9 (Fig.10). The PS combination '11141-2222-121' yielded the highest correlation, and '11131-2222-111' showed the highest performance according to Taylor's score.

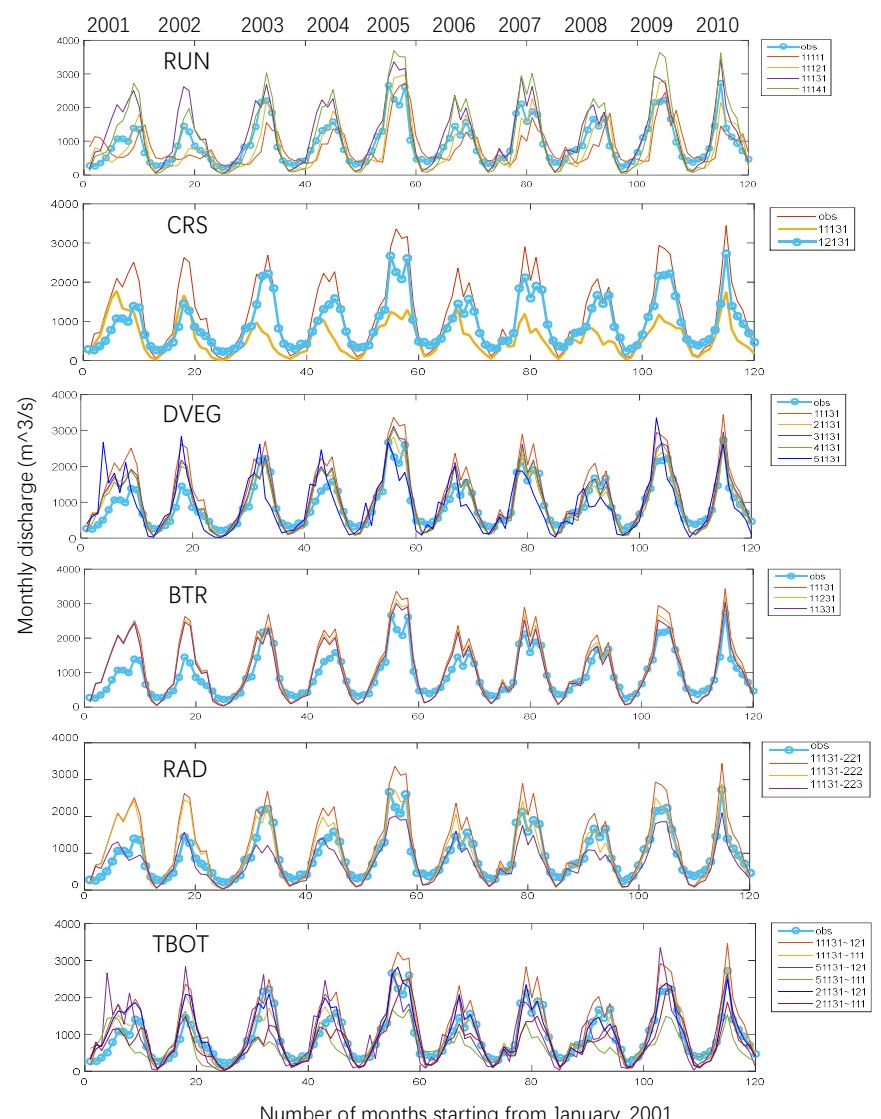

491

**Figure 9:** Simulated monthly river discharge (m³/s) for Lanzhou by NMH-CS. The first subplot

displays the results simulated with varying RUN schemes while the other subplots follow a similar

pattern. Reconstructed natural discharge is denoted as 'obs'.

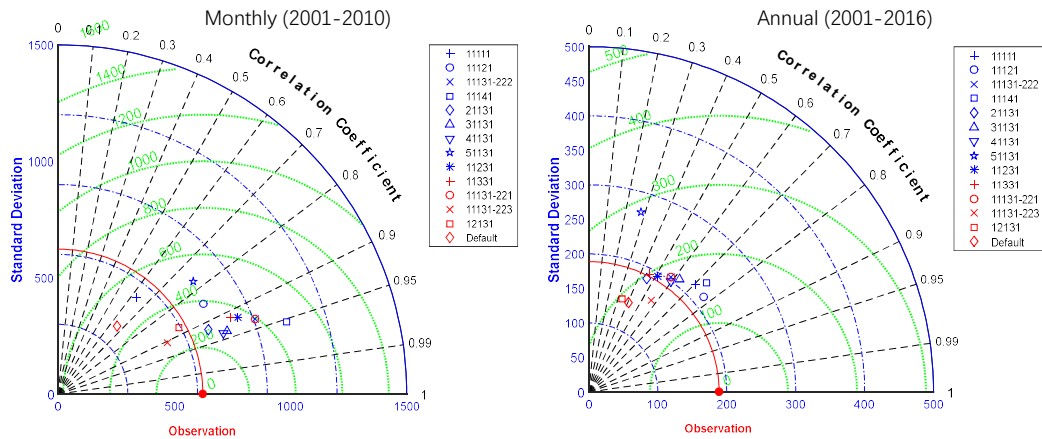

495

**Figure 10:** Taylor diagram for monthly and annual mean river discharge (m³/s) at the Lanzhou

monitoring station

**Table 4**: Performance of various parameterization schemes on monthly discharge for Lanzhou station

| | scheme | Winter | summer |
|---|---|---|---|
| RUN | 1.SIGGM | Overestimation | underestimation |
| | 2.SIMTOP | | Small overestimation |
| | 3.Schaake96 | underestimation | Large overestimation |
| | 4.BATS | | The Largest overestimation |
| CRS | 1. Ball-Berry | Small difference | Overestimation |
| | 2. Jarvis | | Underestimation |
| DVEG | 1.Table LAI, read FVEG | | The largest overestimation |
| | 2.dynamical LAI and FVEG=f(LAI) | No significant difference | Mostly small overestimation |
| | 3. table LAI, FVEG=f(LAI) | | |
| | 4 table LAI, FVEG=maximum | | |
| | 5.Dynamical LAI, maximum FVEG | Unstable overestimation and underestimation | |
| BTR | 1.Noah | No significant difference | The largest overestimation |
| | 2.CLM | | The middle overestimation |
| | 3.SSib | | The smallest overestimation |
| RAD | 1. gap=F(3D, cosz) | No significant difference | Large overestimation |
| | 2. gap=0 | | Slight overestimation |
| | 3. gap=1-FVEG | | underestimation |
| TBOT | 1. Zero flux | No significant difference | large |
| | 2.Noah | | small |

## 5 Discussions

### 5.1 Major advantages of NMH-CS

The original intention of developing a Noah-MP model with the C# programming language was to analyze and edit Noah-MP code in a more efficient way, as there are many modern and efficient tools available for analyzing code written in C#, such as Microsoft Visual Studio, SharpDevelop (https://github.com/icsharpcode/SharpDevelop). There are almost no comparable powerful tools for analyzing FORTRAN code. This advantage is significant from the developers' perspective.

From the user's perspective, NMH-CS run on windows (although it should also run on other UNIX like platforms after some specific configuration in the future), which is more favorable for many Windows users around the world. On the windows system, in most cases, the NMH-CS software can be distributed at multiple computers by simply copying it, unlike that in Unix-like systems, compiling of the code is usually required.

### 5.3 Inconsistencies between the two models

As indicated by the previous analysis, the main inconsistency between the outputs of the two models (WRF-Hydro and NMH-CS) was found to be related to the transition between the presence or absence

of snow layers. In Noah-MP, the existence of snow layers is determined by the depth of the snow (represented by the variable SNOWH in the code). When the SNOWH value approaches the threshold (0.025m), a small error will result in a division on the judgement whether a snow layer exists. This inconsistent division will further lead to significant differences in other state variables. However, this difference will not last long (as shown in Figure 3, up to 30-40 days). However, it is difficult to determine whether the errors in SNOWH are caused by the accumulation of floating-point errors or other errors.

There may be some other inconsistencies that has not been identified. Due to the complex nature of Noah-MP, it is challenging to identify all the minor differences through the process of code checking and debugging. Therefore, to ensure the results of two models to be completely consistent need a long-term process. Discrepancies can be arisen from multiple factors, including floating-point calculation errors, some inconsistent hardcoded parameter values (as local variables in certain modules), or inconsistent programming code. The former two are reasonable and acceptable, whereas a coding mismatch can cause unexpected outputs. From the scientific perspective, these minor differences between NMH-CS and WRF Hydro are not very critical, as the model users are always modifying the code during their research, and small changes in the code can lead to large different results. The existence of differences does not always mean that NMH-CS is inferior to Noah-MP in WRF-Hydro 3.0.

In most cases, identifying discrepancies is only feasible during the debugging of the first 1-3 timesteps, but not for tens to hundreds of subsequent iterations. It is not uncommon for errors to remain undetected even after the execution of numerous time steps. In this study, given that no code inconsistencies were found after multiple rounds of code checking, it is plausible that floating-point errors related to SNOWH (or other related variables) play a major role in explaining the remaining discrepancies.

Based on the debugging process, we also found that some variables such as TV and TG calculated by the two models always have slight inconsistencies, but they are almost insignificant on a daily or monthly scale. It is highly probable that such inconsistencies arise from the accumulated error caused by the recursive calculation of energy transferring for vegetated and bare land.

# 6 Model code and technical documentation for NMH-CS

We archive, manage, and maintain the NMH-CS at https://github.com/lsucksis/NMH-CS for public access. A technical description was provided at the same site. The original version of the model is also provided at the website of Science Data Bank: https://doi.org/10.57760/sciencedb.16102.

# 7 Conclusions

This study presents the NMH-CS 3.0, which is a reconstructed land surface eco-hydrological model based on Noah-MP. The model was developed by translating the FORTRAN code of Noah-MP (in WRF-Hydro 3.0) into C# and also coupling it with a river routing model. The model has been designed for parallel execution on Windows systems, thereby capitalizing on the multi-core CPUs that are now a standard feature of personal computers. The NMH-CS code has been subjected to rigorous testing to

ensure that it produces results that are as consistent as possible with those of the original WRF-Hydro.
The code is based on the C# language, which facilitates greater user-friendliness and facilitates
modification and expansion.
The development of this software enabled the successful execution of 6-km high-resolution simulations
for a rectangular region covering the Yellow River Basin and North China. These simulations were
conducted with a multitude of parameter scheme (PS) combinations within the Noah-MP framework.
Maps of all the outputs (runoff, evaporation, groundwater, energy, vegetation) across the grid domain
demonstrate consistent spatial patterns that are simulated by the two models. The long-term variations
of multiple state variables simulated by the two models also exhibit high consistency, although some
differences also exist. By enabling the coupled river routing modelh, the river discharge simulated by
NMH-CS 3.0 based on the multiple scheme combination of parameterizations is found to be in
reasonable agreement with the reconstructed natural river discharge, for the Lanzhou hydrological
station.
The main inconsistencies in multiple variables between NMH-CS output and WRF-Hydro output was
found to be related to inconsistent judgments on the presence of snow layers, which are caused by
minor cumulative errors near the threshold value of 0.025m for snow depth. Overall, while there are
occasional disparities between the models' outputs, it reproduces highly consistent spatiotemporal
distribution of multiple variables. It can therefore be asserted that NMH-CS can be considered a
reliable replica of Noah-MP in the uncoupled WRF-Hydro 3.0.
This new software NMH-CS can run on Windows system platforms. Its C# code can be analyzed and
visually browsed using many modern intelligent tools such as those in Sharpdevelop or Microsoft Visual
Studio. This feature makes the code easier to analyze and modify, which in turn will attract more users
and promote the future development of the Noah-MP model. The current version of NMH-CS can serve
as a good model for simulating land surface processes in climate change and ecohydrology research.
Although NMH-CS cannot be used as a coupling module to other FORTRAN based framework models
(such as the WRF model), it can still be used as a prototype system to improve the Noah-MP schemes.
Any new improvements in NMH-CS can easily be updated to other FORTRAN based Noah-MP.
Future plans for the development of NMH-CS include (1) providing a single-column run mode and
incorporating a genetic algorithm-based parameter optimization module; (2) extending the functionality
for modelling dynamic vegetation by designing new schemes or optimizing parameters; (3)
implementing major improved model physics that exists in later versions (for example the Noah-MP 5.0)
of Noah-MP into the NMH-CS framework; (4) enabling the functionality of running on UNIX-like
systems.

*Acknowledgements.* Thanks Pei-Rong Lin for her great efforts in guiding Yong-He Liu through the
intricacies of Noah-MP.

*Code and data availability.* 1. The NMH-CS model code is available at
https://doi.org/10.57760/sciencedb.16102 (Liu, 2024a). 2. The Noah-MP technical documentation is
available at the same site and more details will continue to be added in the documentation. 3. The
benchmark meteorological datasets for driving NMH-CS and WRF-Hydro 3.0 were uploaded to the
Science Data Bank (DOI: https://doi.org/10.57760/sciencedb.13122 (Liu, 2024b)).

*Author contributions.* Yong-He Liu has translated the code of WRF-Hydro/Noah-MP to NMH-CS, the

debugging and the benchmark model simulations. The work is led by Zong-Liang Yang. Liu has drafted the paper, with improvement made by Yang.

*Competing interests.* The contact author has declared that none of the authors has any competing interests.

*Financial support.* This study is jointly funded by the "Double First Class" Discipline Construction Project of Surveying and Mapping Science and Technology at Henan University of Technology (GCCYJ202418); Henan Provincial Natural Science Foundation Project (No. 252000421467) and Henan Provincial Key Science and Technology Project (No. 252102320004).

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
