# Peer review of "NMH-CS 3.0: a C# Programming Language and Windows System based Ecohydrological Model Derived from Noah-MP"

_Geoscientific Model Development, 2024_

## Referee Comment (RC2)

Review of *NMP-Hydro 1.0: a C# language and Windows System based Ecohydrological Model Derived from Noah-MP*

Recommendation: Major revisions

The authors have more-or-less replicated the Noah-MP model physics in a C# environment in an effort to expand the Noah-MP and WRF-Hydro research community to personal computers with Windows operating systems. This undertaking is generally reasonable, and a windows ready C# version of Noah-MP coupled with streamflow may be useful to hydro-researchers unfamiliar with Fortran and Linux based computing systems. Further, the results show overall "good enough" agreement between the legacy model and the replication to support/justify use of NMP-Hydro as a research tool. However, there are significant unexplained differences between the two model frameworks that the authors disregard with minor speculation. The paper would be strengthened substantially if the authors actually tracked down the source of these differences and at least documented it as opposed to simply guessing that they are caused by "precision" differences. Such an effort could involve more isolated evaluation of the model components that seem to create these issues. Additionally, the paper would benefit from a cursory "speed" comparison between the WRF-Hydro version of Noah-MP and the NMP-Hydro version, such a comparison would help bolster the motivation for reproducing Noah-MP in C# beyond that of simply "some people don't like Fortran and Unix." Overall, I recommend major revisions with a focus on identifying and discussing why certain model components do not behave exactly as they do in the Fortran environment, and on benchmarking model performance.

**General comments:**

I think the authors should strongly consider a different name for the tool than NMP-Hydro since this is extremely close to WRF-Hydro or NWM branding and is essentially a replica of WRF-Hydro system. This would help differentiate the two modeling systems and avoid confusion within the research community. Perhaps something that involves the C#, since that is the main novelty of the system presented here.

In the experimental configuration, the authors describe using a 6km grid for Noah-MP with a 1 degree meteorological forcing. Is there any meteorological downscaling performed within either WRF-Hydro or NMP-Hydro to reconcile these resolution differences? If not, the simulations would effectively be running ~400 single-column Noah-MP runs with nearly identical meteorological inputs, and the only spatial-detail finer than 1 degree would come from differences in soil texture and land cover class. Further, some of the differences seen between the models, particularly in the winter season could be related to differences in downscaling, so I think it's important to at least clarify whether or not downscaling is applied.

It's not that surprising to me that the largest differences occur during the winter, though I suspect this has relatively little to do with the snow/frozen soil model physics, and rather

may have to do with minor differences in the energy balance over snow that causes small differences in snow temperature to affect snow and surface albedo which can feedback into the energy balance and cause greater model divergence.

Finally, is there any plan to maintain this version of Noah-MP to match new release versions (e.g., Noah-MP 5.0) as the Noah-MP developers at NCAR continue to expand the model capabilities? Even if the authors, justifiably, did not put effort into translating the new code structure associated with version 5.0, is there a plan to try and implement improved model physics into the NMP-Hydro framework? Some discussion around this topic may improve the manuscript.

**Specific Comments:**

**Lines 57 – 59:** This sentence is somewhat misleading and mildly incorrect. True, NoahMP has been integrated seamlessly into the WRF model as a two-way coupled LSM to the atmospheric model and has been since its initial release ~2011/2012. However, WRF-hydro is related to WRF only insofar as it uses the same coding conventions and architecture and can be coupled into the WRF framework. WRF-Hydro itself is a system designed to couple atmospheric forcing to a distributed version of Noah-MP (or other LSMs) and routing/stream flow models. Consider rewording.

**Lines 68 – 70:** additional wording concerns, specifically the words "coupled" and "models" in this context. HRLDAS and WRF-Hydro are not necessarily considered "models" so much they are frameworks used to couple various models together. Also, if you are going to mention the HRLDAS as a framework, consider also including the Land Information System (LIS) here. Consider rewording something like:

*"Noah-MP is supported by several different modeling architectures and frameworks to facilitate coupling it to various other Earth system modeling components including, WRF, MPAS, HRLDAS, LIS, and WRF-Hydro. This makes NoahMP a powerful research and forecasting tool within the hydrology community"*

**Lines 70-72:** This sentence is almost identical to a sentence on lines 59-60, please remove it.

**Line 162:** What is "each variable?" Soil temperature? Snow? Soil moisture? Surface temperature? Energy balance?

**Line 208:** The word "comprising" should be "comprised"

**Line 261:** Please change "*significant differences was*" to "*significant differences were*"

**Lines 288 – 289:** What is meant by "disparate parameter configurations?" If I understand this correctly, does this mean that the parameter tables (i.e., MPTABLE.TBL, etc) that define

specific snow/soil/vegetation properties might be different between WRF-Hydro and NMP-Hydro?  This would be a huge issue trying to validate one against the other.

**Line 296 – 273:** It seems unlikely to me as well that floating-point errors would result in large differences here, LSM's are tightly constrained by the forcing, unlike global atmospheric models for example, such that floating-point errors don't really cascade.  The authors mention snow/frozen soil as potential reasons for the discrepancy, have you looked at snow/frozen soil variables related to runoff?  For example, differences in snow melt, soil ice content or total soil moisture?  The Noah-MP snow and frozen soil models are simple enough that I would not expect trouble when converting code over from one language to another.

**Table 3:** Please change "The first experiment" to "control"

**Lines 311-312**: This line indicates that the authors compared NMP-Hydro with WRF-Hydro, but it's unclear to me that there is a model fun with WRF-Hydro in the experiment suite, rather it looks like a basic parameter-sensitivity study.  Are the authors able to clarify which experiment is run with WRF-Hydro, or are all of the simulations presented in figures 9 and 10 NMP-Hydro?  If that is the case, please edit this line to reflect that so there is no confusion.

**Line 382:** As I understand it here, the modeling isn't "Based on Noah-MP". It *is* Noah-MP, only recoded in the C# programming language and coupled to a streamflow model.  Please edit to be clear about this point.

---

## Referee Comment (RC3)

**Review of *NMP-Hydro 1.0: a C# language and Windows System based Ecohydrological Model Derived from Noah-MP*** (https://doi.org/10.5194/gmd-2024-168)

This manuscript describes a version of Noah-MP that has been ported to C# for the purpose of increased user friendliness and efficiency in model development and testing. This version of Noah-MP is consistent with that used in WRF-Hydro 3.0. The authors refer to this new C# version of Noah-MP as NMP-Hydro. In addition to Noah-MP, NMP-Hydro includes a river routing module. The authors present results from NMP-Hydro and WRF-Hydro and determine there are numerical differences between them, though the two frameworks have identical physics. The authors say that the source of these differences may be floating-point errors. Additionally, the authors provide a comparison of NMP-Hydro discharge and observations from a station within the domain. In this comparison, the authors include many configurations of NMP-Hydro with different physics options activated.

I believe this study addresses an important need for more accessible land surface modeling infrastructures, and I would like to see a revised version that addresses the comments below and those from other reviewers. I think there are some major points that need to be addressed before the manuscript is ready for publication.

Specific comments

**Lines 43-44:** To provide more context, could the authors expand on the ways in which C# is widely used? Also, who is the intended community of users for NMP-Hydro? I ask because many existing users of Noah-MP/WRF-Hydro are comfortable with using these models in Unix/Linux operating systems. Do the authors anticipate that some existing users of Noah-MP/WRF-Hydro will take advantage of the portability and convenience of NMP-Hydro? Do they expect that NMP-Hydro will allow a new community of users to use tools that have traditionally been used by hydrologists and atmospheric scientists? I think widening the accessibility of modeling tools is an important motivation for this work and should be highlighted more in the Introduction.

**Lines 53-54:** Consider replacing "simulation" in these sentences with "component". Using "simulation" is a bit confusing to me. It implies the authors are talking about two different models, but I understand that you are referring to the different components of NMP-Hydro.

**Lines 70-72:** The sentence "Additionally, Noah-MP plays a pivotal role in the National Water Model…" can be omitted, since there is a similar sentence in the previous paragraph.

**Table 1 caption:** I suggest including information here on how the reader can access the Noah-MP user document (or referring them to another part of the manuscript with these details).

**Table 1:** As was suggested by another reviewer, please clarify that Table 1 does not reflect the options currently available in HRLDAS Noah-MP, which many readers will likely be familiar with.

**Table 1:** I suggest elaborating on the scheme options somewhat, as simply "SIMGM", "SIMTOP", "Koren99", "NYO6", "BATS", etc. may not be very informative for a reader who does not have extensive experience with Noah-MP. The authors don't have to completely explain them, but maybe at least say what the acronyms are referring to and include citations for relevant papers, e.g. "Koren's iteration (Koren et al. 1999)". See Table 1 in He et al. (2023) (also published in GMD) for an example of what I mean.

**Line 89:** If possible, can the authors include the version number of the Noah-MP version that was ported to C#? This will help the reader understand how it compares to the current community version (5.0).

**Section 3.2:** I think this section lacks technical detail of the river routing module. In particular, the four contributions from the authors listed in lines 132-135 need elaboration. What are the scientific bases behind these techniques? What is the motivation for their development? Also, can the authors include a figure to summarize the physics of the river routing module? Please add these details to the text or point the reader to the relevant references.

**Section 3.3:** I think this section could also use more detail. How long were the simulations used to check for bugs in the code? Was debugging done based on output from one grid cell within the larger domain? Please add these details to the text.

**Line 198:** Should Fig. 3 be referenced here instead of Fig. 2?

**Line 210:** As was also pointed out by another reviewer, I ask the authors to address the difference in spatial resolution between the GLDAS-1 product (1 degree, quite coarse) and the model simulations (6 km).

**Line 239:** Why were these grid boxes selected for analysis? Please add to the text.

**Table 2:** Please provide a description for CHLEAF in the table.

**Line 246:** Should Table 2 be referenced here instead of Table 3?

**Lines 246-247:** I ask the authors to elaborate on why output for 10 June of different years was chosen for analysis. Why 10 June, and why these particular years? Please consider adding this to the text.

**Line 248:** Mention all of the representative variables included in Figs. 4 and 6 here, not just SFCRNOFF and TV.

**Discussion of Fig. 4, lines 248-253:** I find Fig. 4 to be somewhat misleading. From 4a, 4b, 4e, and 4f, it would seem that there are no visual differences between WRF-Hydro and NMP-Hydro. However, 4c, 4d, 4g, and 4h suggest that there are relative differences of up to 40%, which suggests considerable differences between the two models. Why are such large differences not suggested by 4a, 4b, 4e, and 4f?

I also recommend rearranging the figures such that they are referenced in numerical order, i.e. move Fig. 6 to Fig. 5, move Fig. 8 to Fig. 6, etc. Also move the corresponding discussions as necessary.

**Line 262:** Is the figure reference referring to both Figures 4 and 5?

**Figure 5:** Please revise 5b so the right y-axis labels are fully visible.

**Line 279:** Does NSE refer to Nash-Sutcliffe Efficiency or something else? Please define in the text.

**Figure 6 caption:** Should the units of vegetation temperature be deg C and not K?
**Figure 8:** Please add units to the axes of all subfigures.

**Line 311:** Does this section analyze results from NMP-Hydro and WRF-Hydro, or only NMP-Hydro? It seems Figs. 9 and 10 only include results from NMP-Hydro, but perhaps I am mistaken.

**Lines 385-386:** I don't yet agree that the NMP-Hydro and WRF-Hydro results are consistent. They may be scientifically consistent, but not numerically consistent. I ask that the authors include this distinction in the text.

I agree with another reviewer's comment that the authors should consider renaming NMP-Hydro to something more distinct from WRF-Hydro and Noah-MP to avoid confusion.

**Revised supplementary material:** Perhaps this has already been done in the revised manuscript, but if not, I ask that the authors discuss the supplementary figures in the main text where appropriate.

Technical comments

**Line 16 (abstract) and lines 176-177:** For clarity, change "the most part of North China" to "most of North China"

**Line 235:** Change "percentive" to "percent"

---

## Author Comment (AC2)

**Support Material**

**NMP-Hydro 1.0: a C# language and Windows System based Ecohydrological Model Derived from Noah-MP**

Yong-He Liu[1], Zong-Liang Yang[2]

[1] School of Resources and Environment, Henan Polytechnic University, Jiaozuo, Henan, China

[2] Jackson School of Geoscience, University of Texas at Austin, Austin TX, USA

**Correspondence:** Yong-He Liu (yonghe_hpu@163.com)

[Figure]

**Fig. S1** Comparison of soil water content (SOIL_W, %), soil temperature (SOIL_T, ℃) at different soil layers, snow depth (SNOWH, m) and snow water equivalent (SNEQV, kg m$^{-2}$) between the two models (NMP-Hydro 1.0 versus WRF-Hydro 3.0). The samples are daily values at Gridbox 3 (a sampling position in the grid domain), in 2000-2010.

[Figure]

**Fig.S2** Maps of soil temperature (℃) at the top layer and the lowest layer (the fourth layer) simulated by NMP-Hydro 1.0 and WRF-Hydro 3.0 (a, b, d, e), and the soil temperature differences(℃) between the two models(c,f), at Jan. 1, 2008. The labels for horizontal axis and vertical axis are row numbers and column numbers of the grid domain respectively.

[Figure]

**Fig.S3** Similar to Fig.S2, at Jan. 1, 2008, but for soil water content, represented in volume percent.

[Figure]

**Fig.S4** Similar to Fig.S2 and Fig.S3, at Jan. 1, 2008, but for snow depth (SNOWH, m)(a,b,c) and snow water equivalent(kg·m⁻²)(d,e,f).

[Figure]

**Fig.S5** The daily time series (2000-2010) of soil water content and soil temperature at different grid boxes of some soil layers (not all grid boxes and layers are shown here), simulated by NMP-Hydro (in red) and WRF-Hydro (in blue). The labels for the vertical axis represent the value of soil water content or soil temperature, while the labels for the horizontal axis represent the number of days

starting from 1 January 2000.

[Figure]

**Fig.S6** Similar to Fig.S5, but for snow thickness (m) and snow water equivalent.

---

## Author Comment (AC4)

Dear Referee #1,

Thank you for providing good comments to improve the manuscript.

Specific comments:

1. Is there any statistics for the number of users of C language vs FORTRAN language to highlight the need of developing a C-based model system?

**Reply:** Here, "C language" is a misspelling, while C# language is correct. According to the TIOBE Programming Community Index (www.tiobe.com) for October 2024, C# is the fifth most popular of all major programming languages, with 5.6% of users, while FORTRAN is the ninth most popular, with 1.8% of users.

After our careful thinking, these statistics will not be present in the manuscript, because the number of the language users is not the main reason to reconstruct the model. The main reason is that C# is a modern and powerful language, and is easy to use (by taking advantage of many powerful code analysis tools for C#), while FORTRAN is a traditional old-style language, which is more difficult to use for many users, due to limited code analysis tools.

2. Table 1 caption and related text in the manuscript: please clarify that this is the Noah-MP model version in WRF-Hydro v3.0 not the latest community Noah-MP model version.

**Reply:** The caption is now changed to "Figure 1. The architectural diagram of NMP-Hydro (a) and the conversion of FORTRAN arrays to C# arrays (b). NMP-Hydro is a reconstructed replica of the version of Noah-MP that is coupled in WRF-Hydro 3.0. "

The related text in the manuscript is also minorly modified now. Generally, the original description has mentioned the version.

3. Does the river routing model have to run at the same spatial and temporal resolution as the main Noah-MP column model?

**Reply:** according to this comment, we have added following paragraph in section 3.2:

"*This module requires two additional inputs files, a river segment list file named 'ChannelOrder.txt' and a 'namelist.txt' file. The latter file is used to set parameters and the length of time step. Each river segment in the list file presents following information: its own index, the index of its next downstream river segment, the row number and the column number of the grid box (in Noah-MP's running domain) providing runoff input to the current segment, the length (m) of the current river segment, the two parameter values (K and X) of the Muskingum method, the area of the catchment of the current segment. Each river segment upstream of other segments must be listed ahead of all its downstream segments. The river segment list can be derived from both gridded river network or vectorized river network. The resolution of the river routing is determined by the original river network from which river segment list is derived. Therefore, the choice of using vector river network or gridded river network and the selection of spatial resolution are completely determined by the river segment list file which is provided by the users. The length of the temporal step of the river routing is required to be multiple times shorter than the time step for running the Noah-MP, and can also be designated by the users. In our application, the time step of routing is set to 600s or 900s, while the time step for Noah-MP LSM is set to 3 hours. "*

This description clearly states that the spatial resolution of the river routing model can be determined by the river segment list file. And the time step can be set by users. Therefore, both the spatial and temporal resolution can be very different to the Noah-MP column model.

4. What are the required input data for the river routing model in addition to those needed by Noah-MP column model?

Reply: "This module requires two additional inputs files, a river segment list file named 'ChannelOrder.txt' and a 'namelist.txt' file for setting parameters." This sentence is in the original manuscript.

5. What are the reasons for the difference between the two models? Theoretically speaking, they should produce exactly the same results due to the same equations and input data.

**Reply:** We agree with your comment that the models should produce exactly the same results due to the same equations and input data. However, the equations must be implemented by the programming code. The difference between the C# platform and the Fortran platform is complex, while the Noah-MP model is not a simple one but a model with tens thousands of lines, which made it very difficult for us to discern where the difference is coming from. Fortunately, during this revision, based a hard work on analyze the printed variable values, we find two major inconsistences in the code, and then a large part of the output differences is now removed. **The first previous inconsistence is missing the updating of the 'FICEOLD' variable in each time step; The second inconsistence is the wrong parameter translation of SNOWH2O function, due to the ambiguous parameter defining in the original FORTRAN code. See the improvement in Fig.1.**

[Figure]

**Fig.1 The comparison between NMP-Hydro and WRF-Hydro before (the top four panels) and after (the bottom for panels) this update of NMP-Hydro.**

6. What is the difference in the computational efficiency between the C-based and FORTRAN-based model codes?

**Reply:** We must admit that the program developed by C# language (not the C language in the above misspelling) is usually slower than that developed by Fortran, however C# is not a slow language. Such comparison between the two languages can be seen in many benchmark

comparisons on the internet.

According to our experience, for running a time step of non-parallel WRF-Hydro (the original Noah-MP model) and the non-parallel NMP-Hydro (our newly developed model) on our laptops, the former seems taking less time than the latter. But the parallel running of NMP-Hydro is faster than a non-parallel WRF-Hydro. However, benchmarking comparison is difficult for us to made because the two models usually running on different platforms (operation systems) and computers.

Pursuing higher computational efficiency is not in our goal for reconstructing the Noah-MP model. Therefore, we will not present much discussion on this topic. For most cases, the computational efficiency is not a critical issue, because the difference is always small and acceptable.

7. There are a few important state variables that were not compared between the two models, including soil moisture, soil temperature, snow water equivalent, and snow depth.

**Reply:** There are indeed many important state variables need to concern. **In this revision, we have provided the comparison of soil water content, soil temperature, snow water equivalent and snow depth in supplementary material.** These variables generally show similar effects as the variables that are presented in the main manuscript. Corresponding description was now added to the manuscript.

Due to our limited energy, we cannot provide the tests on all of them everywhere. Consider that these variables are interconnected, the benchmark differences for other variables actually can be indirectly reflected by the results of those presented variables.

8. Lines 250-264: What are the causes for these large differences in runoff, temperature, radiation, and exchange coefficient?

**Reply:** During this revision, we find two code inconsistences through artificial analyzation on the printed variable values, which removed most part of the differences. However, there is still some large differences, which is related to the difference between the states of no-snow layer and having-snow layer. This snow-layer difference may be caused by some accumulated floating-point error on the depth of snow.

9. Lines 271-273: These differences seem too large for precision errors. Usually, if it is single precision, they would only differ in the 7th digit after the decimal point. Did the authors see any difference between the two models for all output fields in the first few (e.g., 10) model timesteps? If not, then maybe the precision error growth in a longer-period run would contribute to such difference. What model timestep did the authors use? Would reducing the model timestep (i.e., smaller numerical integration errors) lead to more consistency between the two models?

**Reply:** Thanks for your recommendation to solve the difference issue of the two models. However, actually, during our efforts of more than five years (Dec. 2018 to now (Sep. 2024)), we have compared the two models in step-by-step running on multiple grid-boxes, only in the first 3-time steps. We find that mostly the differences are very small. We found that there were significant calculation errors (small but cannot be regarded as wrong) after multiple iterations

in the VEG-FLUX function (especially in TV and TG calculations), but due to the iteration, it was difficult to figure out whether the error came from because there were many variables in the function and the code was lengthy and iterative. However, according to the final comparison of output results, in fact, both TV and TG have very small errors between the two models.

All the LSM simulation is based on a 3-hourly timestep. We do not believe that reducing the model time step can get better consistency, because the Noah-MP is not a partial differential equation-based model (unlike that in climate models) and there is no numerical integration concept here.

Fortunately, during this revision, based a hard work on analyze the printed variable values, we find two major inconsistences in the code, and then a large part of the output differences is now removed. **The first previous inconsistence is missing the updating of the 'FICEOLD' variable in each time step; The second inconsistence is the wrong parameter translation of SNOWH2O function, due to the ambiguous parameter defining in the original FORTRAN code.**

10. Is there two-way feedback between Noah-MP and the river routing scheme, or is it just Noah-MP affecting river routing results? Did the authors also see difference between the two Noah-MP models without activating river routing scheme?

**Reply:** There is no feedback from the river routing module to the Noah-MP LSM, therefore, the difference is unrelated to the river routing module.

11. It would be helpful if the authors could discuss a bit the future plans for applying and/or further improving the NMP-Hydro model and potential connection to the broader Noah-MP community.

**Reply:** According to this recommendation, a new paragraph was added to the 'conclusion' section:

"This new software can *run on Windows system platforms. Its C# code can be analyzed and visually browsed using many modern intelligent tools such as those in Sharpdevelop (https://github.com/icsharpcode/SharpDevelop) or Microsoft Visual Studio. The code of NMP-Hydro is easier to analyze, study and modify, which in turn will attract more users and promote the future development of the Noah-MP model. The current version of NMP-Hydro can serve as a good model for simulating land surface processes in climate change and ecohydrology research. Although NMP-Hydro cannot be coupled with the WRF model, it can still be used as a prototype system of Noah-MP to improve the Noah-MP schemes. Any new improvements in NMP-Hydro can easily be updated to other FORTRAN based Noah-MP. Future plans for the development of NMP-Hydro include (1) investigating whether the remaining differences between NMP-Hydro and the original WRF-Hydro 3.0 are caused by floating-point errors or other bugs in the code; (2) providing a single-column run mode and incorporating a genetic algorithm-based parameter optimization module; (3) extending the functionality for modelling dynamic vegetation by designing new schemes or optimizing parameters."*

We have revised the manuscript in a deeper extent, according to your comment.

Here, we upload a supplement file, but the revised manuscript is not allowed to upload here (this is required by the system. I do not understand why). Maybe we can upload the revised manuscript in a later chance.
Thank you very much.

Yonghe Liu

---

## Author Comment (AC5)

Dear Referee #2,

Thank you for your reviewing and giving us so many useful advices. I appreciate your efforts. Now I put my response to your comments in following paragraphs.

Yonghe
2024-12-6

**The comments of Referee #2**

Review of NMP-Hydro 1.0: a C# language and Windows System based Ecohydrological Model Derived from Noah-MP
Recommendation: Major revisions
The authors have more-or-less replicated the Noah-MP model physics in a C# environment in an effort to expand the Noah-MP and WRF-Hydro research community to personal computers with Windows operating systems. This undertaking is generally reasonable, and a windows ready C# version of Noah-MP coupled with streamflow may be useful to hydro- researchers unfamiliar with Fortran and Linux based computing systems. Further, the results show overall "good enough" agreement between the legacy model and the replication to support/justify use of NMP-Hydro as a research tool.
**Reply:** Thank you for acknowledging our work.

However, there are significant unexplained differences between the two model frameworks that the authors disregard with minor speculation. The paper would be strengthened substantially if the authors actually tracked down the source of these differences and at least documented it as opposed to simply guessing that they are caused by "precision" differences. Such an effort could involve more isolated evaluation of the model components that seem to create these issues.
**Reply:** I understand your doubts. It is difficult to determine the actual source of some minor differences. As a model developer with over 16 years of experience, I have extensive knowledge of various types of programming code and know how to debug and identify errors or exceptions. If I can successfully identify the sources of these differences, I will not let them linger until the submission of the manuscript. My development of NMP-Hydro began in the January of 2018, and it has been six years now. During the past six years, I spent at least three months every year in discovering a large number of bugs and resolving code differences, which required me to put in a lot of hard work.
In the manuscript, we mentioned that we performed code validation using breakpoint debugging with the original WRF-Hydro model (although FORTRAN language does not have an efficient breakpoint debugging functionality, we can only use the print clause to print out the variable values). However, such debugging can only be completed in less than 3 time steps. In fact, after only one time step, "floating-point precision" causes small differences between the two models for some variable values, which poses great challenges.
As you suggested, 'isolated evaluation of the model components' is needed, but this suggestion is usually difficult to implement. In Noah-MP's code, there are indeed many functions divided, but these functions are nested with many complex functions. These functions are not pluggable

modules, but rather have many relations between various variables across multiple functions. It is possible to perform error detection within the first running step by isolating different functions, but it is very difficult to perform after running hundreds or thousands of steps.

Fortunately, recently I conducted a new analysis by printing out certain identical variables in WRF-Hydro and our NMP-Hydro. It was only after running for tens months that I find the first significant inconsistency. I finally found a source of error: it was not in the code of the Noah-MP LSM, but at the entrance for calling Noah-MP LSM. Our previous code of NMP-Hydro does not support the real-time updates of the 'FICEOLD'. Now we can ensure that the 'FICEOLD' in our NMP-Hydro is updated in every running step, which eliminates many inconsistencies (Fig.1). Another source of error was identified: a mistranslation of the function 'SNOWH2O' regarding its parameters. The definition of this function in the original FORTRAN code contains several ambiguous points.

Unfortunately, a few inconsistencies remain. The specific source of these discrepancies is associated with the transition between the no-snow layer and the presence of a snow layer. The differences in the simulated output do not consistently manifest. For instance, beginning on January 1, 2000, the first significant error emerged in February 2001; however, when we initiated simulations from 2001 onward, this notable error did not occur.

[Figure]

**Fig.1** The comparison of variables simulated by WRF-Hydro and NMP-Hydro. The upper four: before the correction on the FICEOLD updating in NMP-Hydro; the lower four: after the correction on the FICEOLD updating.

Additionally, the paper would benefit from a cursory "speed" comparison between the WRF-Hydro version of Noah-MP and the NMP-Hydro version, such a comparison would help bolster the motivation for reproducing Noah-MP in C# beyond that of simply "some people don't like Fortran and Unix."

**Reply:** This comparison of the speed between the two models is also difficult to implement. I cannot use the two environments on the same computer, because WRF-Hydro usually runs in a Linux environment, while NMP-Hydro runs in a windows environment. Although my computer uses the windows 11 which supports Linux system (the WSL system), but I failed to correctly compile the code of WRF-Hydro on my WSL system.

I also do not think that users are very interested in comparison of running speed between the two models. Everyone knows that FORTRAN and C can run faster than C#, because

FORTRAN/C is a relatively low-level language compared to any modern object-oriented computer language. However, as a language that can run in native machine code, C# is not a slow one. The speed comparison of C# and Fortran (or C) can be found in many documents on the internet, therefore, there is no need to compare here. The difference in the two models' speed is mainly governed by the difference between the two languages. According to my experience (comparison between two different computers), WRF-Hydro indeed runs faster than NMP-Hydro when the latter runs in a non-parallel mode. NMP-Hydro can run in a parallel mode on a personal computer, while WRF-Hydro cannot. WRF-Hydro can run in a parallel mode using a MPI environment of high-performance computers, while NMP-Hydro does not support any MPI environment.

As for the speed comparison with different threads used, for NMP-Hydro, we have presented some tests in this revision:" *We tested the time it takes for NMP-Hydro to execute by setting multiple C# parallel threads. The computer used for the testing is a common laptop with 6 CPU cores. The results indicate that for the execution of the entire domain, as the number of threads increases from 1 to 6, the average time consumed for each time step is 1576ms, 977ms, 801ms, 711ms, 679ms, and 672ms, respectively. When the number of threads is set to 1, the time consumption is slightly greater than the time for the execution in the non- parallel mode (1461ms). It is worth noting that the time spent is not linearly related to the number of parallel threads, which can be explained by various reasons. One is that some tasks are actually not executed in parallel mode, such as reading meteorological input files. Another reason is that not all threads in NMP-Hydro are fully processed by the CPU core, as there are many other tasks in the entire Windows environment that have to be processed simultaneously by the same CPU cores.*"

Overall, I recommend major revisions with a focus on identifying and discussing why certain model components do not behave exactly as they do in the Fortran environment, and on benchmarking model performance.

**Reply:** I also aim to diligently minimize any discrepancies between the two models prior to drafting the manuscript. However, I have already invested more 6 years in addressing these issues, and there are indeed some inconsistencies in the output. I find it quite challenging to completely resolve these problems based on my own strength. Furthermore, the Noah-MP model cannot be easily decomposed into multiple components as imagined, because many variables are intricately interconnected across multiple 'functions'. For instance, a minor variation in a specific variable can lead to significant inconsistencies after several time steps, making it difficult to identify the source of such discrepancies. From a technical perspective, capturing all subtle differences across different languages poses great challenges.

Fortunately, I have identified two major sources of the differences and resolved them now.

**General comments:**

I think the authors should strongly consider a different name for the tool than NMP-Hydro since this is extremely close to WRF-Hydro or NWM branding and is essentially a replica of WRF-Hydro system. This would help differentiate the two modeling systems and avoid confusion within the research community. Perhaps something that involves the C#, since that is the main novelty of the system presented here.

**Reply:** Considering that two reviewers, including you, believe that renaming is necessary, I now decide to rename it during the upload phase of this revision. Renaming the model is not an easy task, as the code has already been submitted to multiple sites. In this revision, we must use the original name.

In fact, I don't think the current name NMP-Hydro is very similar to WRF-Hydro or NWM. After all, many brand names nowadays are similar and there is no standard to avoid similarities. If we successfully change it to another name, will other reviewers or readers also have different opinions?

In the experimental configuration, the authors describe using a 6km grid for Noah-MP with a 1 degree meteorological forcing. Is there any meteorological downscaling performed within either WRF-Hydro or NMP-Hydro to reconcile these resolution differences If not, the simulations would effectively be running ~400 single-column Noah-MP runs with nearly identical meteorological inputs, and the only spatial-detail finer than 1 degree would come from differences in soil texture and land cover class. Further, some of the differences seen between the models, particularly in the winter season could be related to differences in downscaling, so I think it's important to at least clarify whether or not downscaling is applied.

**Reply:** The manuscript now describes this information in a clearer way. No downscaling was used here, and the 6-kilometer resolution of the driving dataset was only the result of regridding using bilinear interpolation. Whether to use downscaling is actually unrelated to the comparison. I just need to ensure that these two models use the same forced dataset (both downscaled and regridded data are acceptable).

It's not that surprising to me that the largest differences occur during the winter, though I suspect this has relatively little to do with the snow/frozen soil model physics, and rather may have to do with minor differences in the energy balance over snow that causes small differences in snow temperature to affect snow and surface albedo which can feedback into the energy balance and cause greater model divergence.

**Reply:** My analysis supports your viewpoint. In this revision, I found an important code inconsistency: the FICEOLD variable was not updated at each time step. The correction of FICEOLD update has eliminated a large number of inconsistencies, but there are still some inconsistencies. Although I cannot guarantee that these remainder inconsistencies are related to errors in the snow, based on code analysis, they are likely caused by the accumulation of inconsistencies in the snow (variables SNOWH and DZSNSO). However, it is difficult to find the reasons for the differences in snow variables, as many variables have small differences, and the growth of differences in snow variables seems to be moderate. It is highly probable that the differences in SNOWH is due to some local accumulation of floating-point errors.

In the code of Noah-MP, if SNOWH (the thickness of snow) is less than 0.025m, the ISNOW is zero (means no snow layer), otherwise, the ISNOW is 1 (means there is one snow layer). **Fig.**2 shows that the inconsistences in SNOWH cause different ISNOW (the layer number of top snow layer) shifts. The major inconsistences in snow equivalent (SNEQV), vegetation temperature (TV), the depth of the top snow layer (DZSNSO) are related to the inconsistences of ISNOW. This can explain that when there is a snow layer, it will give rise to different TV (TG (ground temperature) as well) and SNEQV values.

[Figure]

Fig.2 The differences (WRF-Hydro minus NMP-Hydro) between the variables simulated by NMP-Hydro and WRF-Hydro. The plots are based on three-hourly series, starting from 1 January, 2000.

Finally, is there any plan to maintain this version of Noah-MP to match new release versions (e.g., Noah-MP 5.0) as the Noah-MP developers at NCAR continue to expand the model capabilities. Even if the authors, justifiably, did not put effort into translating the new code structure associated with version 5.0, is there a plan to try and implement improved model physics into the NMP-Hydro framework. Some discussion around this topic may improve the manuscript.

**Reply:** Implementing new model physics into the model is necessary. However, I am the only developer for the development of NMP-Hydro and the task has spent more than 6 years. Testing new models by reference to a legacy model is a very time-consuming task and is not as easy as someone imagined. Therefore, I will not consider bigger plans in the future because it seems unrealistic for me now. In my opinion, there are always some uncertainties in any module of

physics, continuously adding more modules may not bring additional benefits to the scientific studies.

In this revision, I have added some plans which need to address in the future.

Specific Comments:

Lines 57 – 59: This sentence is somewhat misleading and mildly incorrect. True, NoahMP has been integrated seamlessly into the WRF model as a two-way coupled LSM to the atmospheric model and has been since its initial release ~2011/2012. However, WRF-hydro is related to WRF only insofar as it uses the same coding conventions and architecture and can be coupled into the WRF framework. WRF-Hydro itself is a system designed to couple atmospheric forcing to a distributed version of Noah-MP (or other LSMs) and routing/stream flow models. Consider rewording.

**Reply:** Thank you for pointing out the wrong description. I am not familiar with the deep details of these models. Now, I modified this description: *Based on Noah-MP, WRF-Hydro was developed, and can be seamlessly integrated into the Weather Research and Forecasting (WRF) model. (Gochis, 2020). Furthermore, the offline WRF-Hydro model plays a pivotal role in the National Water Model, contributing to the simulation of floods and river flows across the United States.*

Lines 68 – 70: additional wording concerns, specifically the words "coupled" and "models" in this context. HRLDAS and WRF-Hydro are not necessarily considered "models" so much they are frameworks used to couple various models together. Also, if you are going to mention the HRLDAS as a framework, consider also including the Land Information System (LIS) here. Consider rewording something like:

"Noah-MP is supported by several diGerent modeling architectures and frameworks to facilitate coupling it to various other Earth system modeling components including, WRF, MPAS, HRLDAS, LIS, and WRF-Hydro. This makes NoahMP a powerful research and forecasting tool within the hydrology community"

**Reply:** Based on your good advice. I modified the text accordingly.

Lines 70-72: This sentence is almost identical to a sentence on lines 59-60, please remove it.

**Reply: Removed now.**

Line 162: What is "each variable"? Soil temperature? Snow? Soil moisture? Surface Temperature? Energy balance?

**Reply:** Here the variables refer to all the variables in the programming code. Not only the physical variables as you mentioned, but many local variables that can influence the simulation. Now, the code is rewritten as "*Initially, the code underwent a meticulous step-by-step check by examining the printed values of many variables (including many local variables in the code) in WRF-Hydro 3.0 running for specific single columns.*"

Line 208: The word "comprising" should be "comprised"

**Reply: Corrected.**

Line 261: Please change "significant differences was" to "significant differences were"

**Reply: Corrected.**

Lines 288 – 289: What is meant by "disparate parameter configurations?" If I understand this correctly, does this mean that the parameter tables (i.e., MPTABLE.TBL, etc) that define specific snow/soil/vegetation properties might be different between WRF-Hydro and NMP-Hydro? This would be a huge issue trying to validate one against the other.

**Reply:** It refers to the hardcoded local parameters (usually local variables in a certain function). There are many of them. I have changed the sentence:" *Such discrepancies may be attributed to a number of factors, including floating-point calculation errors, some inconsistent hardcoded parameter values (as local variables in certain functions), or encoding inconsistencies.*"

Line 296 – 273: It seems unlikely to me as well that floating-point errors would result in large differences here, LSM's are tightly constrained by the forcing, unlike global atmospheric models for example, such that floating-point errors don't really cascade. The authors mention snow/frozen soil as potential reasons for the discrepancy, have you looked at snow/frozen soil variables related to runoff. For example, differences in snow melt, soil ice content or total soil moisture. The Noah-MP snow and frozen soil models are simple enough that I would not expect trouble when converting code over from one language to another.

**Reply:** Thank you for telling me that "LSM's are tightly constrained by the forcing". During this revision, I have made deeper trace on many variables. Although two major sources of difference were discerned and resolved now, but there are still differences left. These differences also are found to be related to the transition between no-snow layer and the presence of a snow layer. The difference in snow layer transition is caused by the differences in SNOWH, however, what causes the difference in SNOWH is difficult to resolve now. Generally, these differences do not cascade over many time steps.

Table 3: Please change "The first experiment" to "control"

**Reply:** Modified.

Lines 311-312: This line indicates that the authors compared NMP-Hydro with WRF-Hydro, but it's unclear to me that there is a model fun with WRF-Hydro in the experiment suite, rather it looks like a basic parameter-sensitivity study. Are the authors able to clarify which experiment is run with WRF-Hydro, or are all of the simulations presented in figures 9 and 10 NMP-Hydro? If that is the case, please edit this line to reflect that so there is no confusion.

**Reply:** This is a wrong description in the previous manuscript. No more comparison between the two models is presented in this section. The sentence is modified as "Here, we present the numerical outputs of NMP-Hydro on the streamflow discharges over the Yellow River, with various parameterization schemes were used."

Line 382: As I understand it here, the modeling isn't "Based on Noah-MP". It is Noah-MP, only recoded in the C# programming language and coupled to a streamflow model. Please edit to be clear about this point.

**Reply:** Your understanding is correct. We have made some revision on the descriptions over the entire manuscript.

---

## Author Comment (AC6)

Dear Referee #3,

Thank you for providing good comments to improve the manuscript. We are now preparing the revision of the manuscript based on your comments.

**Review of *NMP-Hydro 1.0: a C# language and Windows System based Ecohydrological Model Derived from Noah-MP*** (https://doi.org/10.5194/gmd-2024-168)

This manuscript describes a version of Noah-MP that has been ported to C# for the purpose of increased user friendliness and efficiency in model development and testing. This version of Noah-MP is consistent with that used in WRF-Hydro 3.0. The authors refer to this new C# version of Noah-MP as NMP-Hydro. In addition to Noah-MP, NMP-Hydro includes a river routing module. The authors present results from NMP-Hydro and WRF-Hydro and determine there are numerical differences between them, though the two frameworks have identical physics. The authors say that the source of these differences may be floating-point errors. Additionally, the authors provide a comparison of NMP-Hydro discharge and observations from a station within the domain. In this comparison, the authors include many configurations of NMP-Hydro with different physics options activated.

I believe this study addresses an important need for more accessible land surface modeling infrastructures, and I would like to see a revised version that addresses the comments below and those from other reviewers. I think there are some major points that need to be addressed before the manuscript is ready for publication.

**Reply:** Thank you for acknowledging our work.

Specific comments

**Lines 43-44:** To provide more context, could the authors expand on the ways in which C# is widely used? Also, who is the intended community of users for NMP-Hydro? I ask because many existing users of Noah-MP/WRF-Hydro are comfortable with using these models in Unix/Linux operating systems. Do the authors anticipate that some existing users of Noah-MP/WRF-Hydro will take advantage of the portability and convenience of NMP-Hydro? Do they expect that NMP-Hydro will allow a new community of users to use tools that have traditionally been used by hydrologists and atmospheric scientists? I think widening the accessibility of modeling tools is an important motivation for this work and should be highlighted more in the Introduction.

**Reply:** This comment is interesting. As a modern programming language, C# is extensively utilized by numerous commercial enterprises in industrial software development and scientific research. According to the TIOBE Programming Community Index for October 2024, C# ranks fifth among major programming languages with a user base of 5.6%, while Fortran ranks ninth with 1.8% of users. It is important to note that the languages preceding C# in popularity are Python, C++, Java, and C.

I believe that some existing users of Noah-MP may be reluctant to transition their working environment to the Windows system. However, many new hydrological researchers, primarily postgraduate students, are not proficient with Linux. They are currently compelled to use Linux because the existing Noah-MP is developed in Fortran under this operating system. If a Windows-based version of Noah-MP were available, it would likely be more appealing to these new researchers. For existing users who are already familiar with Linux, a C#-based Noah-MP could also be attractive. As a modern programming language, C# offers greater user-friendliness, ease of

use, and access to more powerful development tools compared to Fortran. We will present some description in the introduction, according to your comment.

**Lines 53-54:** Consider replacing "simulation" in these sentences with "component". Using "simulation" is a bit confusing to me. It implies the authors are talking about two different models, but I understand that you are referring to the different components of NMP-Hydro.
Reply: Modified now.

**Lines 70-72:** The sentence "Additionally, Noah-MP plays a pivotal role in the National Water Model…" can be omitted, since there is a similar sentence in the previous paragraph.
Reply: Corrected.

**Table 1 caption:** I suggest including information here on how the reader can access the Noah-MP user document (or referring them to another part of the manuscript with these details).
Reply: Now cited to "Gochis, D.J., W. Yu, D.N. Yates, 2015: The WRF-Hydro model technical description and user's guide, version 3.0. NCAR Technical Document. 123 pages."

**Table 1:** As was suggested by another reviewer, please clarify that Table 1 does not reflect the options currently available in HRLDAS Noah-MP, which many readers will likely be familiar with.
**Reply: A sentence is added in the caption: *Note that these options may not be applicable to other versions of Noah-MP, such as that used in HRLDAS*.**

**Table 1:** I suggest elaborating on the scheme options somewhat, as simply "SIMGM", "SIMTOP", "Koren99", "NYO6", "BATS", etc. may not be very informative for a reader who does not have extensive experience with Noah-MP. The authors don't have to completely explain them, but maybe at least say what the acronyms are referring to and include citations for relevant papers, e.g. "Koren's iteration (Koren et al. 1999)". See Table 1 in He et al. (2023) (also published in GMD) for an example of what I mean.
**Reply:** We think elaborating these marks is unnecessary here, because these marks/acronyms are used in the namelist file of the original Noah-MP, and why the earlier developers use them is not very clear to us. Most existing Noah-MP users are familiar with these options. If the users want to know the actual meaning of these acronyms, they should read the document of the original Noah-MP other than the description of this study. In other word, we are not responsible for explaining the initial details of Noah-MP.

**Line 89:** If possible, can the authors include the version number of the Noah-MP version that was ported to C#? This will help the reader understand how it compares to the current community version (5.0).
**Reply:** Based on your comments, we have changed the version of NMP-Hydro from 1.0 to 3.0, in order to maintain consistency with the original WRF-Hydro 3.0.

**Section 3.2:** I think this section lacks technical detail of the river routing module. In particular, the four contributions from the authors listed in lines 132-135 need elaboration. What are the scientific bases behind these techniques? What is the motivation for them development? Also, can the authors include a figure to summarize the physics of the river routing module? Please add these details to the text or point the reader to the relevant references.

**Reply: The original description here is not very clear, now some small modification has been made.** The river routing module is actually a previously published study (Liu et al., 2023), where the details of the module can be found. There is no much physics in the Muskingum-method river routing. The principle is very simple, because the Muskingum method is a traditionally widely used and known one. The main innovation of this module is the new parallelization method other than the physics.

**Section 3.3:** I think this section could also use more detail. How long were the simulations used to check for bugs in the code? Was debugging done based on output from one grid cell within the larger domain? Please add these details to the text.

Reply: Now the total paragraph is rewritten and more details are described.

**Line 198:** Should Fig. 3 be referenced here instead of Fig. 2?

Reply: Yes, Fig.3 is correct. Modified.

**Line 210:** As was also pointed out by another reviewer, I ask the authors to address the difference in spatial resolution between the GLDAS-1 product (1 degree, quite coarse) and the model simulations (6 km).

**Reply:** More details are added now : the data extraction include spatial clipping and regridding using bilinear interpolation.

**Line 239:** Why were these grid boxes selected for analysis? Please add to the text.

**Reply: More details are added now.** The selection of these grid points is an arbitrary determination by roughly considering different climate zones, without strict considerations. In fact, in this study, more grid points have been tested, but the results are mostly similar. Here, only analyzes and discusses the results based on these three grid points.

**Table 2:** Please provide a description for CHLEAF in the table.

Reply: Added.

**Line 246:** Should Table 2 be referenced here instead of Table 3?

Reply: Yes, corrected.

**Lines 246-247:** I ask the authors to elaborate on why output for 10 June of different years was chosen for analysis. Why 10 June, and why these particular years? Please consider adding this to the text.

**Reply:** The time slices are arbitrarily selected without special consideration. After all, the amount of data is very large, and it is impossible to display the results of all time data. Only a few dates can

be selected. I think it is unnecessary to describe every reason for the selection. If no special reason is given in the text, then it must be that this reason is not important.

**Line 248:** Mention all of the representative variables included in Figs. 4 and 6 here, not just SFCRNOFF and TV.

Reply: Has been rewritten:" *The maps for all the sate variables in Table 2 reflect high consistence between NMP-Hydro and WRF-Hydro, but here only the maps for two representative variables (SFCRNOFF and TV) are shown in Fig.4 and Fig.6.*"

There are numerous variables are compared in our study, it is difficult to present all the results in such figures, considering that the length of the paper for publication must be limited. The figures in Figs.4-6 are enough and representative, because other variables are highly interrelated with SFCRNOFF and TV. If there are inconsistencies in other variable, it is impossible get such inconsistent SFCRNOFF and TV maps.

**Discussion of Fig. 4, lines 248-253:** I find Fig. 4 to be somewhat misleading. From 4a, 4b, 4e, and 4f, it would seem that there are no visual differences between WRF-Hydro and NMP-Hydro. However, 4c, 4d, 4g, and 4h suggest that there are relative differences of up to 40%, which suggests considerable differences between the two models. Why are such large differences not suggested by 4a, 4b, 4e, and 4f?

I also recommend rearranging the figures such that they are referenced in numerical order, i.e. move Fig. 6 to Fig. 5, move Fig. 8 to Fig. 6, etc. Also move the corresponding discussions as necessary.

**Reply:** Thank you for pointing out the problem. We will rewrite this discussion during the submission of this revision.

**Line 262:** Is the figure reference referring to both Figures 4 and 5?

Reply: The original description is ambiguous. We need rewrite the description in the total paragraph, because the result is now changed a lot.

**Figure 5:** Please revise 5b so the right y-axis labels are fully visible.

Reply: Revised the figure.

**Line 279:** Does NSE refer to Nash-Sutcliffe Efficiency or something else? Please define in the text.

Reply: During this revision, the metrics such as R and NSE will no longer be used in the paper, because the variables are more highly consistent now.

**Figure 6 caption:** Should the units of vegetation temperature be deg C and not K?

Reply: During this revision, All K will be changed to deg C. the figures will be redrawn.

**Figure 8:** Please add units to the axes of all subfigures.

Reply: Now units are added. The figures will be reproduced.

**Line 311:** Does this section analyze results from NMP-Hydro and WRF-Hydro, or only NMP-

Hydro?

It seems Figs. 9 and 10 only include results from NMP-Hydro, but perhaps I am mistaken.

**Reply:** No results of WRF-Hydro is presented here (some description is wrong in the previous submission). The purpose of this section is to test the effectiveness of NMP-Hydro, other than compare the two of them. Due to the many physical scheme combinations, conducting both the two models with all those scheme combinations are difficult and also is not very necessary.

**Lines 385-386:** I don't yet agree that the NMP-Hydro and WRF-Hydro results are consistent. They may be scientifically consistent, but not numerically consistent. I ask that the authors include this distinction in the text.

**Reply:** The sentence has been changed to " The NMP-Hydro code has been subjected to rigorous testing to ensure that it produces results that are as consistent as possible with those of the original WRF-Hydro."

I agree with another reviewer's comment that the authors should consider renaming NMPHydro to something more distinct from WRF-Hydro and Noah-MP to avoid confusion.

**Reply:** I have considered your suggestion, and I also think the current name is not ideally good. However, in this revision, I have not come up with a more suitable name for it. Meanwhile, renaming it now poses some difficulties as it involves changing the names on several websites. If multiple reviewers agree that a name change is necessary, I will decide to do so in the next upload of the manuscript.

Actually, I don't think NMP-Hydro would be confused with WRF-Hydro or Noah-MP, as they are different spells. The reason I choose such a name because this model does come from WRF-Hydro or Noah-MP.

**Revised supplementary material:** Perhaps this has already been done in the revised manuscript, but if not, I ask that the authors discuss the supplementary figures in the main text where appropriate.

**Reply:** I would discuss those figures as far as possible.

Technical comments

**Line 16 (abstract) and lines 176-177:** For clarity, change "the most part of North China" to "most of North China"

Reply: Corrected.

**Line 235:** Change "percentive" to "percent"

Reply: Corrected.

**Your suggestions are very valuable. We feel very appreciated for your hard working.**

**The authors: Yong-He and Zong-Liang**

**References:**

Liu, Y., Yang, Z. and Lin, P., 2023. Parallel river channel routing computation based on a straightforward domain decomposition of river networks. JOURNAL OF HYDROLOGY, 625.

---

## Referee Report (RR1)

Overall, I think the authors have done a great job revising this manuscript. I am satisfied with the responses to my previous comments. I have some other (mostly small) suggestions below. Once these comments are addressed, I think the manuscript will be suitable for publication. I would be happy to review the revised manuscript again.

Lines 32-33: Citations missing.

Line 35: FORTRAN is misspelled.

Line 56: 'Analyse' should be analysing.

Line 89: Remove 'are'.

Line 141: Change 'simulating' to 'simulate'.

Lines 153-154: Here, n and n+1 are used to refer to different time steps, while in Eq. 2, t and t+1 are used.

Line 207: Citations missing.

Line 212: Citation missing.

Line 248: Remove 'of'.

Line 261: Citation missing, and parentheses need to be fixed.

Line 362: Can the authors comment on possible factors that could be contributing to these sporadic differences in high-latitude areas? Is this related to the discussion of Fig. 7?

Fig. 4 & 5: Can these figures be plotted on a latitude-longitude grid? Just to provide a geographic reference in terms of where the larger differences are occurring. The same goes for similar figures in the supplementary info.

Fig 4: Can the authors comment further on the region of higher relative difference in underground runoff shown by the red in the lower right figure?

Line 375: Should the "Noah-MP" in the caption here be omitted?

Fig 9: Can the authors plot these figures using the same y-axis scale for all plots? This will aid in comparison between plots.

Line 551: 6 km is the simulation resolution, not the total size of the simulation domain. Please clarify this.

Line 556: Remove 'on river network' to make this sentence clearer.

Line 563: Remove 'happened'.

Lines 562-565: It's not necessary to refer to WRF-Hydro twice in the sentence, and I recommend re-wording it for clarity.

---

## Author Response (AR2)

**The second round of revision on the manuscript**

**"NMH-CS 3.0: a C# Programming Language and Windows System based Ecohydrological Model Derived from Noah-MP"**

Dear referees and editors,

Thank you for identifying the issues in the manuscript. We have made modifications based on your feedback.

The following paragraph presents the point-to-point response to your comments.

Yonghe Liu

Overall, I think the authors have done a great job revising this manuscript. I am satisfied with the responses to my previous comments. I have some other (mostly small) suggestions below. Once these comments are addressed, I think the manuscript will be suitable for publication. I would be happy to review the revised manuscript again.
**Reply:** Thank you for recognizing our work.

Lines 32-33: Citations missing.
**Reply:** I have not made clear which citation is missing. Only citations (Niu et al., 2011; Yang et al., 2011) are here, which are listed in the references.

Line 35: FORTRAN is misspelled.
**Reply:** Corrected.

Line 56: 'Analyse' should be analysing.
**Reply:** Corrected.

Line 89: Remove 'are'.
**Reply:** Corrected.

Line 141: Change 'simulating' to 'simulate'.
**Reply:** Corrected.

Lines 153-154: Here, n and n+1 are used to refer to diUerent time steps, while in Eq. 2, t and t+1 are used.
**Reply:** Corrected. All the 'n' are replaced by 't'

Line 207: Citations missing.
**Reply:** Added now: (David et al., 2015; Mizukami et al., 2016), in the reference list.

Line 212: Citation missing.
**Reply:** Added now, same literature to the above.

Line 248: Remove 'of'.
**Reply:** Removed.

Line 261: Citation missing, and parentheses need to be fixed.
**Reply:** Added now: (David et al., 2013) in the reference list.

Line 362: Can the authors comment on possible factors that could be contributing to these sporadic differences in high-latitude areas? Is this related to the discussion of Fig. 7?
**Reply:** We cannot certify whether the factors contributing to these differences are identical to those discussed in relation to Fig. 7. However, this is highly probable. Now, I added a sentence to the discussion of Fig.7 in Line 404: "This explanation may account for the inconsistencies observed in the time series in Fig. 7 and the sporadically distributed discrepancies in high-latitude regions depicted in Fig. 4."

Fig. 4 & 5: Can these figures be plotted on a latitude-longitude grid? Just to provide a geographic reference in terms of where the larger differences are occurring. The same goes for similar figures in the supplementary info.
**Reply:** The data is in a Lambert projection coordinate system, so it is difficult in plotting the maps in a latitude-longitude due to the lack of powerful tools. Now I have added the latitude-longitude network on the maps.

Fig 4: Can the authors comment further on the region of higher relative difference in underground runoff shown by the red in the lower right figure?
**Reply:** We cannot make clear what is the reason of such differences in the current study. Although this relative difference is significant in the map, actually it is actually in very smalls because the annual total underground runoff in the corresponding region is below 50 mm. We think this may be a floating-point error. We added one sentence in line 359: "These regions with large relative differences of underground runoff actually are in small absolute differences, primarily because the annual total groundwater runoff in these areas is inherently low (<50 mm). This discrepancy is likely attributable to floating-point arithmetic errors. "

Line 375: Should the "Noah-MP" in the caption here be omitted?
**Reply:** Noah-MP is omitted now.

Fig 9: Can the authors plot these figures using the same y-axis scale for all plots? This will aid in comparison between plots.
**Reply:** I have redrawn the curves now using the similar y-axis scale. It is difficult to present all the curves in one single plot (see following attached plot). Actually, the comparison between different subplots is also unnecessary and irrelevant to the topic under discussion here. When the If readers want to compare the different performances of option combinations, they can see

the Taylor diagrams in the manuscript.

[Figure]

**An attached plot for all the curves together**

Line 551: 6 km is the simulation resolution, not the total size of the simulation domain. Please clarify this.

**Reply:** It is a wrong description. Now it is corrected. 6 km refers to the resolution.

Line 556: Remove 'on river network' to make this sentence clearer.

**Reply:** Removed.

Line 563: Remove 'happened'.

**Reply:** Removed.

Lines 562-565: It's not necessary to refer to WRF-Hydro twice in the sentence, and I recommend re-wording it for clarity.

**Reply:** The sentences are rewritten: "Overall, while there are occasional disparities between the models' outputs, it reproduces highly consistent spatiotemporal distribution of multiple variables."

**References**

David, C.H., Famiglietti, J.S., Yang, Z. and Eijkhout, V.: Enhanced fixed-size parallel speedup with the Muskingum method using a trans-boundary approach and a large subbasins approximation, Water Resour. Res., 51(9), 7547-7571, https://doi.org/10.1002/2014WR016650, 2015.

David, C.H., Yang, Z. and Famiglietti, J.S.: Quantification of the upstream-to-downstream influence in the Muskingum method and implications for speedup in parallel computations of river flow, Water Resour. Res., 49(5), 2783-2800, https://doi.org/10.1002/wrcr.20250, 2013.

Mizukami, N., Clark, M.P., Sampson, K., Nijssen, B., Mao, Y., McMillan, H., Viger, R.J., Markstrom, S.L., Hay, L.E., Woods, R., Arnold, J.R. and Brekke, L.D.: mizuRoute version 1: a river network routing tool for a continental domain water resources applications, Geosci Model Dev, 9(6), 2223-2238, https://doi.org/10.5194/gmd-9-2223-2016, 2016.

Niu, G., Yang, Z., Mitchell, K.E., Chen, F., Ek, M.B., Barlage, M., Kumar, A., Manning, K., Niyogi, D., Rosero, E., Tewari, M. and Xia, Y.: The community Noah land surface model with

multiparameterization options (Noah-MP): 1. Model description and evaluation with local-scale measurements, J. Geophys. Res.-Atmos., 116(D12109), https://doi.org/10.1029/2010JD015139, 2011.

Yang, Z., Niu, G., Mitchell, K.E., Chen, F., Ek, M.B., Barlage, M., Longuevergne, L., Manning, K., Niyogi, D., Tewari, M. and Xia, Y.: The community Noah land surface model with multiparameterization options (Noah-MP): 2. Evaluation over global river basins, J. Geophys. Res.-Atmos., 116(D12110), https://doi.org/10.1029/2010JD015140, 2011.

---

## Author Response (AR3)

**The final modification on the manuscript**

"NMH-CS 3.0: a C# Programming Language and Windows System based

Ecohydrological Model Derived from Noah-MP"

Dear editors,

    Thank you for accepting our manuscript. There are no more comments from the reviewers. We found that some figures are redundant, so **we updated the original 10 figures to 8 figures now, by merging some of them**.

    We feel appreciated for your great efforts.

Yonghe Liu